

# Technical descriptions of the experimental dynamical downscaling simulations over North America by the CAM5.4-MPAS4.0 variable-resolution model

Koichi Sakaguchi[1], L. Ruby Leung[1], Colin M. Zarzycki[2], Jihyeon Jang[3], Seth McGinnis[4], Bryce E. Harrop[1], William C. Skamarock[3], Andrew Gettelman[5], Chun Zhao[6], William J. Gutowski[7], Stephen Leak[8], and Linda Mearns[4]

[1]Atmospheric Sciences and Global Change Division, Pacific Northwest National Laboratory, Richland, WA, USA
[2]Department of Meteorology and Atmospheric Science, Pennsylvania State University, University Park, PA, USA
[3]Mesoscale and Microscale Meteorology Laboratory, National Center for Atmospheric Research, Boulder, CO, USA
[4]Research Application Laboratory, National Center for Atmospheric Research, Boulder, CO, USA
[5]Climate and Global Dynamics Laboratory, National Center for Atmospheric Research, Boulder, CO, USA
[6]School of Earth and Space Sciences, University of Science and Technology of China, Hefei, Anhui, China
[7]Geological And Atmospheric Sciences Department, Iowa State University, Ames, IA, USA
[8]The National Energy Research Scientific Computing Center, Berkeley, CA, USA

**Correspondence:** Koichi Sakaguchi (Koichi.Sakaguchi@pnnl.gov)

**Abstract.** Comprehensive assessment of climate datasets is important for communicating to stakeholders model projections and associated uncertainties. Uncertainties can arise not only from assumptions and biases within the model but also from external factors such as computational constraint and data processing. To understand sources of uncertainties in global variable-resolution (VR) dynamical downscaling, we produced a regional climate dataset using the Model for Prediction Across Scales

dynamical core version 4.0 coupled to the Community Atmosphere Model version 5.4 (CAM-MPAS). This document provides technical details of the model configuration, simulations, computational requirements, post-processing, and data archive of the experimental CAM-MPAS downscaling data.

The CAM-MPAS model is configured with VR meshes featuring higher resolutions over North America, as well as quasi-uniform resolution meshes across the globe. The dataset includes multiple uniform- (240 and 120 km) and variable-resolution

(50-200, 25-100, and 12-46 km) simulations for both the present-day (1990-2010) and future (2080-2100) periods, closely following the protocol of the North American Coordinated Regional Climate Downscaling Experiment. A deviation from the protocol is the pseudo-warming experiment for the future period, using the ocean boundary conditions produced by adding the sea surface temperature and sea ice changes from the low resolution version of the Max Planck Institute Earth System Model in the Coupled Model Intercomparison Project phase five to the present-day ocean state from a reanalysis product.

Some unique aspects of global VR models are evaluated to provide background knowledge to data users and to explore good practices for modelers who use VR models for regional downscaling. In the coarse-resolution domain, strong resolution-sensitivity of the hydrological cycles exists over the tropics but does not appear to affect the mid-latitude circulations in the Northern Hemisphere including the downscaling target of North America. The pseudo-warming experiment leads to similar responses of large-scale circulations to the imposed radiative and boundary forcings in the CAM-MPAS and MPI models,



but their climatological states in the historical period differ over various regions including North America. Such differences are carried to the future period, suggesting the importance of the base state climatology. Within the refined domain, precipitation statistics improve with higher resolutions, and such statistical inference is verified to be negligibly influenced by horizontal remapping during post-processing. Limited ($\approx 50\%$ slower) throughput of the current code is found on a recent many-core/wide-vector High Performance Computing system, which limits the lengths of the 12-46 km simulations and in-

directly affects the uncertainty from sampling. Our experience shows that global and technical aspects of VR downscaling framework require further investigations to reduce uncertainties for regional refinement.

# 1 Introduction

With increasing frequencies and intensities of extreme events witnessed in the last decades worldwide, there is an increasing need for high-resolution climate information to support risk assessment and climate adaptation and mitigation planning

(Gutowski Jr. et al., 2020). However, limited by computing resources and model structures, climate projections produced by global climate and earth system models, including those in the most recent Coupled Model Intercomparison Project Phase 6 (CMIP6) (Eyring et al., 2016), are mostly available at grid spacing of 100-150 km. These models do not adequately resolve regional climate variability associated with forcing such as mesoscale surface heterogeneities and orography (Roberts et al., 2018). A subset of global models participated in the High Resolution Model Intercomparison Project feature grid spacing be-

tween 25 and 50 km, but the high computational cost leads to smaller ensemble sizes, fewer types of experiments, and shorter simulation lengths than those for the models with standard grid spacing (Haarsma et al., 2016). To bridge the scale gap, diverse statistical and dynamical approaches have been developed to downscale global climate simulations to higher resolutions (4-50 km grid spacing) for different regions around the world (e.g. Wilby and Dawson, 2013; Giorgi and Mearns, 1991; Giorgi and Gutowski, 2015; Prein et al., 2017). These downscaling approaches have been compared to inform methodological develop-

ment and to provide uncertainty information for users of the downscaled climate data (e.g. Wood et al., 2004; Fowler et al., 2007; Trzaska and Schnarr, 2014; Smid and Costa, 2018). However, few attempts (e.g. Wilby et al., 2000) have been made to compare different statistical and dynamical downscaling methods under the same experimental protocol to reduce factors confounding interpretation of the results.

The effort described in this work was initiated in a project supported by the U.S. Department of Energy, "A Hierarchical

Evaluation Framework for Assessing Climate Simulations Relevant to the Energy-Water-Land Nexus (FACETS)", which aims to systematically compare representative dynamical and statistical downscaling methods to evaluate and understand their relative credibility for projecting regional climate change. The project has been expanded to a larger project, "A Framework for Improving Analysis and Modeling of Earth System and Intersectoral Dynamics at Regional Scales (HyperFACETS)", with a larger multi-institutional team (https://hyperfacets.ucdavis.edu/). Through both project stages, we produced a model evalua-

tion framework that features a set of structured, hierarchical experiments performed using different statistical and dynamical downscaling methods and models, and a cascade of metrics informed by the different uses of regional climate information (e.g.



Bukovsky et al., 2017; Rhoades et al., 2018a, b; Pendergrass et al., 2020; Pryor et al., 2020; Pryor and Schoof, 2020; Coburn and Pryor, 2021; Feng et al., 2021).

Dynamical downscaling usually refers to numerical simulations over a limited-area domain to achieve a higher resolution than those of global climate models (e.g. Giorgi and Mearns, 1991; Giorgi, 2019). Outputs from a global model simulation are used to provide the boundary conditions. This one-way nesting approach does not allow interactions between the target high-resolution domain and the rest of the globe, and needs to deal with various issues from the prescribed lateral boundary conditions (Wang et al., 2004). Another dynamical downscaling approach is global variable-resolution (VR) models. A class of VR models uses the so-called stretched grid that is transformed continuously and non-locally to achieve finer grid spacings over a specified region while grid cells are "stretched" (coarsened) in other regions of the global domain, retaining the same number of grid columns (e.g. Fox-Rabinovitz et al., 2000; McGregor, 2013). Several models of this class were compared under the Stretched Grid Model Intercomparison Project (Fox-Rabinovitz et al., 2006). The other class of VR models increases the grid density locally over specified region(s) without a compensating reduction of grid resolution over other parts of the globe. Such a regional refinement is achieved by unstructured grids whose cell distributions are determined to tile the surface of a sphere nearly uniformly, instead of being tied to geographical structures such as latitude and longitude coordinates (Williamson, 2007; Staniforth and Thuburn, 2011; Ju et al., 2011). The regional downscaling dataset described in this study are produced by the latter VR approach.

As a part of the structured hierarchical experiments, we have produced a regional climate dataset using a global variable resolution dynamical core called Model for Prediction Across Scales (MPAS) coupled with the Community Atmosphere Model (CAM) physics suite. The CAM-MPAS model allows high-resolution regional simulations to be performed using regional refinement facilitated by unstructured grids, along with its non-hydrostatic dynamics, climate-oriented CAM physics parameterizations, and other Earth system component models available in the Community Earth System Model (CESM). For the dataset presented here, the model is configured on VR meshes with regional refinement over North America and quasi-uniform resolution (UR) meshes across the globe (Figure 1). The VR configurations allow fine-scale features to be better resolved inside the refinement region, which interact seamlessly with the large-scale circulations simulated at coarser resolution outside the refined domain.

The dataset is designed to be compatible with the regional climate simulations produced for the North American CORDEX program (Mearns et al., 2017) (NA-CORDEX) and additional simulations using the Advanced Research Weather Research and Forecasting (WRF) Model and RegCM4 models conducted under the HyperFACETS project. Few studies have compared limited-area and global VR dynamical downscaling approaches at the climate time scale (Hagos et al., 2013; Huang et al., 2016; Xu et al., 2018, 2021), making such comparisons an important element of the HyperFACETS project. For example, limited-area models are applied to specific regions conditioned on the global model simulated large-scale circulation prescribed through lateral boundary conditions. Their lateral boundary conditions are identical regardless of the resolution of the downscaling grid. In contrast, global variable resolution models simulate both the regional and global climate in a single model. Unlike limited-area models, winds flowing into the regionally refinement domain can vary with the resolutions of the coarse-resolution domains and the transition zones and potentially through the upscale effects from the high-resolution domain. As can be seen



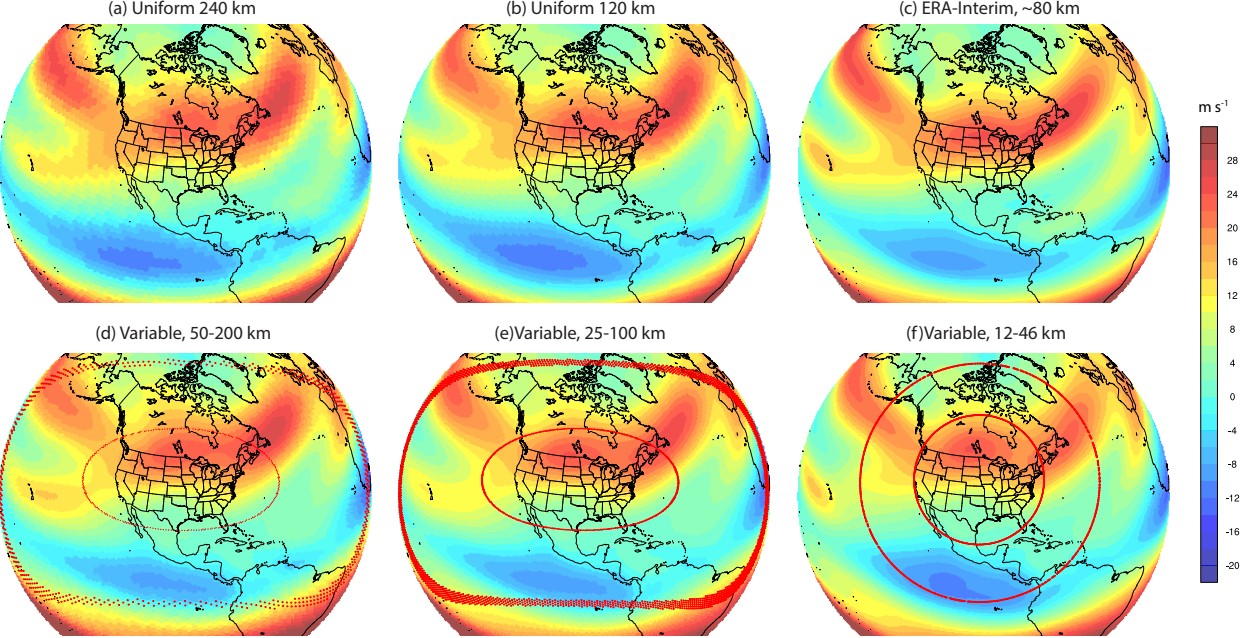

**Figure 1.** JJA-mean zonal wind at the 200 hPa level in each of the present-day (eval) CAM-MPAS simulations and ERA-Interim, whose sea surface temperature and sea ice fraction were used for the ocean surface boundary forcing: (a) globally uniform 240km grid, (b) uniform 120 km, (c) ERA-Interim, (d) variable-resolution grid with 50 km grid spacing over North America and 200 km in the coarse-resolution domain, (e) variable-resolutions from 100 km to 25 km, and (f) variable-resolutions from 46 km to 12 km. In (d)-(f), gridcells at approximate boundaries between the coarse-resolution, transition, and refined domains are marked by red dots.

in Figure 1, the general pattern of the large-scale winds is similar across simulations at different resolutions and ERA-Interim (Dee et al., 2011), which provides the ocean boundary condition to the model. However, the zonal wind pattern in the eastern Pacific near California shows notable sensitivity to resolution (and bias against ERA-Interim), which could affect downwind

regional hydrometeorology.

As a relatively new approach, the VR framework has not been widely used in coordinated downscaling experiments. Therefore potential users of the CAM-MPAS climate dataset are not expected to be familiar with the characteristics of the model and the specificity regarding the model outputs. It is also not clear if one can apply an experimental protocol developed for regional models straightforwardly to global VR models. Furthermore, the timing of our production simulations coincided with

the introduction of new, many-core architectures of the High Performance Computing (HPC) system, such as Cori Knights Landing at the National Energy Research Scientific Computing Center (NERSC). Climate simulations of our CAM-MPAS code on such a system revealed challenges that are relevant to the wider community that conduct global and regional climate simulations. Hence, the goal of this paper is to provide a reference for not only the users of the experimental CAM-MPAS



downscaled climate dataset but also the future users of the CESM2-MPAS and other global VR models for regional downscal-
ing. Specifically, we provide: a technical summary of the CAM-MPAS model (Sect. 2), details of the CAM-MPAS downscaling
experiments (Sect. 3), a description of the post-processing of model outputs and archival (Sect. 4), and general characteristics
of model simulations (Sect. 5).

## 2   Model description

Previous works have already introduced the CAM-MPAS framework (Rauscher et al., 2013; Sakaguchi et al., 2015; Zhao et al.,
2016), but in this section we reiterate the descriptions of the MPAS and CAM models and their coupling for the convenience
of readers and the completeness of this document. More details are available from the cited references.

### 2.1   The Model for Prediction Across Scales (MPAS)

MPAS is a modeling framework developed to simulate geophysical fluid dynamics over a wide range of scales (Skamarock
et al., 2012; Ringler et al., 2013). Currently four models based on the MPAS framework exist: atmosphere, ocean, sea ice, and
land ice (The MPAS project, 2013). The atmosphere-version (MPAS-Atmosphere) solves the compressible, non-hydrostatic
momentum and mass-conservation equations coupled to a thermodynamic energy equation (Skamarock et al., 2012). The novel
characteristic of the MPAS framework is a C-grid finite-volume scheme developed for a hexagonal unstructured grid called
Spherical Centroidal Voronoi Tesselations (SCVT) (Ringler et al., 2010), accompanied by a new scalar transport scheme by
Skamarock and Gassmann (2011). The SCVT mesh can be constructed to have either quasi-uniform grid cell sizes or variable
ones with smooth transitions between the coarse and fine resolution regions (Ju et al., 2011). The C-grid staggering provides
an advantage in resolving divergent flows important to mesoscale features, and the finite-volume formulation guarantees local
conservative property for prognostic variables of the dynamical core (Skamarock et al., 2012). MPAS-Atmosphere is available
as a stand-alone global atmosphere model with its own suite of subgrid parameterizations (Duda et al., 2015), but here we
use the MPAS-Atmosphere numerical solver as the dynamical core coupled to the CAM physics parameterizations. Previous
studies using VR meshes demonstrated that the MPAS dynamical core is able to simulate atmospheric flow across coarse and
fine-resolution regions without unphysical signals (Park et al., 2013; Rauscher et al., 2013). This capability of regional mesh
refinement is the main feature we aim to test in the context of dynamical downscaling for regional climate projections.

### 2.2   The Community Earth System Model version 2 (CESM2) and Community Atmosphere Model version 5.4 (CAM5.4)

The model code base used for our simulations is a beta version of CESM2 (CESM1.5), the same code used by Gettelman et al.
(2018), who focused on regional refinement capability of the spectral element dynamical core. The atmospheric component
model CAM has multiple versions of physics parameterization package. We use the CAM version 5.4, which is an interim
version toward CAM version 6 (Bogenschutz et al., 2018) and was the default physics option for this version of CESM/CAM
code. The parameterization components in CAM5.4 are summarized in Table 1. Their characteristics are documented in detail



by Bogenschutz et al. (2018), and a variety of diagnostic plots are publicly available (Atmosphere Model Working Group, 2015). A major difference between CAM5.4 and the previous version CAM5.0 is the prognostic mass and number concentrations of rain and snow in the new cloud microphysics scheme, MG2 (Gettelman and Morrison, 2015; Gettelman et al., 2015). Prognostic concentrations of precipitating particles make the model more appropriate for high-resolution simulations by removing assumptions necessary for a diagnostic approach (e.g., neglecting the advection of precipitating particles, Rhoades

et al. (2018b)). The prognostic aerosol scheme is also revised as the four-mode version of the Modal Aerosol Module (MAM4) (Liu et al., 2016), but we only use the diagnostic aerosol scheme (Bacmeister et al., 2014) for the simulations documented in this paper. Specifically, the monthly-mean aerosol mass concentrations for the year 2000 are derived from a previous simulation using CAM version 4 with the prognostic three-moment modal aerosol scheme (Liu et al., 2012) on a one-degree grid. Given the prescribed aerosol mass concentrations, aerosol number concentrations are calculated by an empirical relationship between

the two concentrations, and passed to the cloud microphysics.

**Table 1.** Physics parameterizations in CAM5.4

| Process | Reference |
| --- | --- |
| Boundary layer | Bretherton and Park (2009) |
| Cloud macrophysics | Park et al. (2014) |
| Cloud microphysics | Gettelman and Morrison (2015); Gettelman et al. (2015) |
| Deep convection | Zhang and McFarlane (1995); Neale et al. (2008) |
| Shallow convection | Park and Bretherton (2009) |
| Prescribed aerosol | Kiehl et al. (2000); Bacmeister et al. (2014) |
| Radiative transfer | Iacono et al. (2008) |
| Turbulent mountain stress | Richter et al. (2010) |

## 2.3    CAM-MPAS coupling

An early effort to port the MPAS dynamical core to the CESM/CAM model started in 2011 under the project "Development of Frameworks for Robust Regional Climate Modeling" (Leung et al., 2013). The hydrostatic solver of the pre-released version of MPAS (Park et al., 2013) was coupled to CAM4 by the collaborative work among Los Alamos National Laboratory, Lawrence

Livermore National Laboratory, and the National Center for Atmospheric Research. This CAM-MPAS model was extensively evaluated through a hierarchy of experiments (Hagos et al., 2013; Rauscher et al., 2013; Rauscher and Ringler, 2014; Sakaguchi et al., 2015, 2016; Zhao et al., 2016). Those studies demonstrated the ability of VR simulations to reproduce the uniform, globally high-resolution simulations inside the refined domain in terms of the characteristics of atmospheric circulations as well as the sensitivity of the physics parameterizations to horizontal resolution. In the idealized aquaplanet configuration with

the older CAM4 physics, the resolution sensitivity of moist physics leads to unphysical upscale effects (Hagos et al., 2013; Rauscher et al., 2013), but these artifacts are mostly muted when an interactive land model is coupled, along with the presence





of other forcing such as topography and land/ocean contrast (Sakaguchi et al., 2015). The non-hydrostatic version of the MPAS dynamical core (the released version 2) was later coupled to CESM version 1.5 to understand the behavior of the CAM5 physics in a wide range of resolutions over seasonal or longer time scales (Zhao et al., 2016; Hagos et al., 2018). Hagos et al.

(2018) used this model with a convection-permitting VR mesh (32 km to 4km) to study the sensitivity of extreme precipitation to several parameters in the CAM5 physics, demonstrating stable coupling between the non-hydrostatic MPAS dynamical core and the global model physics package CAM5 at kilometer-scale resolution. The CAM-MPAS model for the present work is similar to the one used by Hagos et al. (2018), except that MPAS v2 is replaced by a more recent version 4. The same CAM-MPAS version as used in this study has demonstrated robust performance in simulating the Asian monsoon system using

30-120 km VR mesh (Liang et al., 2021).

The coupling between the non-hydrostatic MPAS and the main driver of CAM uses a Fortran interface and calling sequence similar to the default finite-volume (FV) and other dynamical cores available in CAM (Neale et al., 2010). With this coupling approach, the dynamical core can be switched from the default FV to MPAS by simply providing a flag "CAM_DYCORE=mpas" to the CESM build script (env_build.xml), along with an appropriate name of the horizontal grid

(e.g., "mp120a" has been defined for the UR120 grid following CESM Software Engineering Group (2014)). As shown in Fig. 2, the MPAS dynamical core receives tendencies of horizontal momentum and temperature predicted by physics parameterizations, along with the mixing ratios of tracers. MPAS cycles its time steps from the previous atmospheric state with the physics tendencies used as forcing terms. After MPAS completes its (sub) time steps, the updated atmospheric and tracer states are passed to the CAM physics.

The vertical grid follows the height-based coordinate used by MPAS-Atmosphere (Klemp, 2011), but the number of layers (32) and the height of the interface levels are configured to closely match those of the hybrid $\sigma$-$p$ coordinate used by other CAM dynamical cores. Hydrostatic pressure, geopotential height, and the pressure thickness of each grid box are passed to the CAM physics, which operates on a vertical column under hydrostatic balance. Accordingly, the pressure vertical velocity $\omega$ passed from MPAS to the CAM physics is diagnosed under the hydrostatic balance and is different from the non-hydrostatic vertical

velocity prognostically simulated in the MPAS dynamical core. A second-order diffusion is added to the top three model layers to produce the so-called sponge-layers following other CAM dynamical cores (Jablonowski and Williamson, 2011; Lauritzen et al., 2012, 2018). The model top level is located at about 45 km above the sea level. This model top is higher than those typically used in MPAS-Atmosphere ($\approx$ 30 km). On the other hand, the number of vertical levels in CAM5.4 is smaller than the default vertical levels in the MPAS-Atmosphere (41 in version 4), resulting in a relatively coarse vertical resolution for a

mesoscale model. But its vertical resolution is within the range used by regional models participating in NA-CORDEX (18 to 58 levels across models).

This experimental version of CAM-MPAS is available from our private repository on Github (see the Code and data availability section), but it is not an official release and does not offer the same technical support as other CAM versions. Some model structural differences between CAM and MPAS, such as the vertical coordinate, require further work to improve physical

consistency throughout the coupling processes. An on-going effort to port MPAS to CAM/CESM addresses those remaining



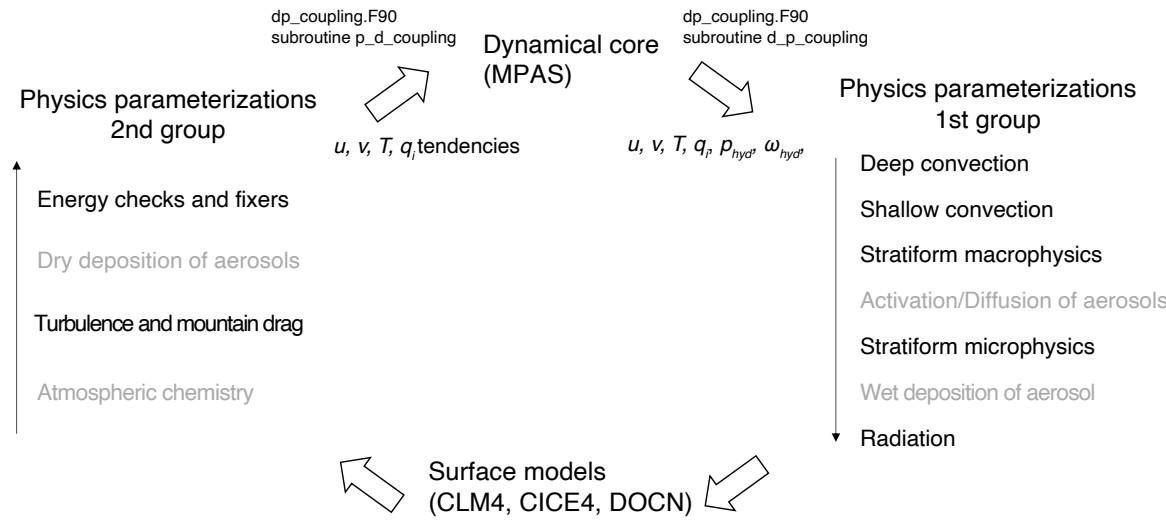

**Figure 2.** Process-coupling sequence in the CAM-MPAS model. The MPAS dynamical core receives the time rate of change of zonal and meridional winds (u, v), atmospheric temperature (T), and water in vapor and condensed phases ($q_i$, with i = 1, 2, 3, ... for water vapor, cloud liquid, cloud ice, etc.) from subgrid parameterizations and returns an updated atmospheric state in terms of u, v, T, $q_i$, hydrostatic pressure ($P_{hyd}$), pressure velocity ($\omega_{hyd}$) obtained through the hydrostatic relationship from the prognostic vertical velocity in MPAS, and geopotential $\Phi$, after integrating adiabatic dynamics. Also shown are the names of the source code file and subroutines where the coupling operations are carried out. The parameterizations shown in gray color were not active in our CAM-MPAS simulations.

technical issues as part of the System for Integrated Modeling of the Atmosphere (SIMA) project (Gettelman et al., 2021; Lauritzen and Truesdale, 2021; Huang et al., 2022).

## 3 CAM-MPAS downscaling experiments

### 3.1 Model configuration

Three VR grids (50-200km, 25-100km, and 12-46km) and two UR grids (240km and 120km) are used for the CAM-MPAS downscaling experiment (Table. 2). Figure 3 illustrates the UR and VR grids and the distributions of grid cell spacing in the three VR grids. The experiment is composed of decadal simulations for the present-day and the end of the twenty-first century under the Representative Concentration Pathway (RCP) 8.5, featuring a business-as-usual scenario leading to a radiative forcing of 8.5 Wm$^{-2}$ by the end of this century. The two simulations are named following the CORDEX project protocol: "eval"

denotes the historical simulations using reanalysis data for boundary conditions for its principal role of model evaluation





against observations. "rcp85" denotes the future simulations in which the external forcings follow the RCP8.5 scenario and the ocean/sea ice boundary conditions are prescribed by adding the GCM-simulated climate change signals to the historical observations, the so-called pseudo-global warming experiment. We selected the Max-Planck-Institute Earth System Model low resolution version (MPI-ESM-LR) (Giorgetta et al., 2013) from the nine GCMs considered in NA-CORDEX based on the model performance of the warm-season precipitation over central U.S. (Sakaguchi et al., 2021). Note that limited area models participating in NA-CORDEX include another historical simulation called "hist", in which the lateral and bottom boundary conditions are provided by the driving global models. Also the rcp85 simulations in NA-CORDEX use GCM outputs directly for boundary conditions ("direct downscaling") in contrast to adding the climate-change signals to the observed present-day boundary conditions. We do not conduct the "hist" experiment with CAM-MPAS because our principal goal is to assess the credibility of dynamically downscaled climate by the CAM-MPAS atmosphere model in comparison to observational and other downscaled data, which at a minimum requires 1) the "eval" run with the prescribed ocean boundary conditions from observations, isolating the CAM-MPAS model's bias without the influence of the GCM's SST and sea ice biases, and 2) model response to external forcings associated with global warming, which can be reasonably assessed by the pseudo-global warming experiment (see the general agreement in the large-scale climate response between the CAM-MPAS and MPI models in Sect. 5.2.2). An advantage of the global VR simulation in pseudo-global warming approach is that, unlike adding the mean atmospheric climate change signals to the lateral boundary conditions for regional models, a global VR simulation does include variability and high-order atmospheric responses to the warming. Because our dataset does not have the "hist" experiment, we will use the terms "eval", "historical", and "present-day" interchangeably to refer to the eval simulations.

The five CAM-MPAS model resolutions are named UR240, UR120, VR50-200, VR25-100, and VR12-46. UR240 has a similar grid spacing as MPI-ESM-LR. The UR120 grid has a comparable resolution to those in the majority of CMIP5 and CMIP6 models. Although their grid spacing does not exactly match those of the coarse-resolution domains on the VR grids nor the MPI-ESM-LR model, these two UR meshes are readily available from the MPAS website and serve as a reference for the VR simulations. The two VR grids, VR50-200 and VR25-100, are created for this project because similar VR grids were not available from the MPAS mesh archive when the project started. The two meshes are designed to have rectangular-shaped high-resolution domain over CONUS (Fig. 3), resembling the regional model domain for NA-CORDEX (CORDEX, 2015). The 12-46km VR mesh is obtained from the MPAS mesh archive and has a circular, and slightly smaller (by ≈30%) high-resolution domain than the other two VR grids, but still covers the most of the NA CORDEX domain.

As the default parameters in the CAM5.4 physics are tuned for the prognostic MAM4 aerosol model, we re-tuned CAM5.4 with the prescribed aerosol and the CAM-default dynamical core, Finite-Volume (FV) scheme, on its nominal one-degree global grid. No attempt has been made to tune model parameters differently for the MPAS dynamical core and at each resolution. While we are aware that resolution-dependent tuning and/or scale-aware physics schemes are necessary to fully take advantage of increased resolution (Bacmeister et al., 2014; Xie et al., 2018), tuning each resolution for both global and regional climate requires extensive effort (e.g. Hourdin et al., 2017) and is left for future work. We also note that resolution-dependent tuning is not usually done for limited-area models participated in NA-CORDEX and HyperFACETS, as well as other coordinated projects that cover multiple model resolutions (e.g. Haarsma et al., 2016). The following parameters, however, are





**Figure 3.** Illustration of the MPAS meshes used for this study: (a) uniform resolution (240km) and (b) variable resolution (50-200km). In (a) the approximate domain for NA-CORDEX experiment is shown by a black line. In (b) approximate boundaries between the 50-km domain and transition zone, and the transition zone and 200-km domain are marked by red markers. The three histograms show the numbers of grid columns binned by grid cell spacing (km) for (c) VR50-200, (d) VR25-100, and (e) VR12-46.

changed for each resolution: timestep lengths, numerical diffusion coefficients, and the convective time scale used in the Zhang-McFarlane deep convection scheme (Table. 3). The dynamics timesteps are initially set as $\triangle t = 6 \times \triangle x$ and further





**Table 2.** List of simulations. The simulation period does not include 1-2 spinup years. Regional grids are used for post-processed data and defined in NA-CORDEX, except for NAM-88i and NAM-176i, which are defined in a similar manner to the other NA-CORDEX grids.

| No. | Name | Model grid | Regional grid (grid spacing) | simulation period |
|-----|------|-----------|------------------------------|-------------------|
| 1 | UR240-eval | Quasi-uniform 240km | NAM-176i (2.0°) | 1990-2010 |
| 2 | UR120-eval | Quasi-uniform 120km | NAM-88i (1.0°) | 1990-2010 |
| 3 | VR50-200-eval | Variable-resolution 50-200km | NAM-44i (0.50°) | 1990-2010 |
| 4 | VR25-100-eval | Variable-resolution 25-100km | NAM-22i (0.25°) | 1990-2010 |
| 5 | VR12-46-eval | Variable-resolution 12-46km | NAM-11i (0.125°) | 2001-2010 |
| 6 | UR240-rcp85 | Quasi-uniform 240km | NAM-176i (2.0°) | 2080-2100 |
| 7 | UR120-rcp85 | Quasi-uniform 120km | NAM-88i (1.0°) | 2080-2100 |
| 8 | VR50-200-rcp85 | Variable-resolution 50-200km | NAM-44i (0.50°) | 2080-2100 |
| 9 | VR25-100-rcp85 | Variable-resolution 25-100km | NAM-22i (0.25°) | 2080-2100 |
| 10 | VR12-46-rcp85 | Variable-resolution 12-46km | NAM-11i (0.125°) | 2091-2100 |

adjusted to avoid numerical instabilities that tend to occur within the stratospheric jet over the Andes. In VR simulations, the dynamics timestep is constrained by the smallest grid spacing in the refined region. The physics timestep is scaled from the
235 default 1800 s for ≈1° grid spacing in the same ratio as grid spacing changes. The convection time scale is then adjusted to scale with the physics timestep in order to reduce sensitivities to horizontal resolution and timestep (Mishra and Srinivasan, 2010; Williamson, 2013; Gross et al., 2018).

**Table 3.** Resolution-dependent parameters. The default physics timestep and convective time scale is 1800 s and 3600 s, respectively.

| Model grid | CAM timestep (s) | MPAS timestep (s) | Convective time scale (s) |
|-----------|------------------|-------------------|---------------------------|
| UR240 | 1800 | 900 | 3600 |
| UR120 | 1800 | 450 | 3600 |
| VR50-200 | 900 | 150 | 1800 |
| VR25-100 | 600 | 85 | 1200 |
| VR12-46 | 300 | 60 | 600 |

We use a predefined CESM component set "FAMIPC5" that automatically configures CESM and its input data (e.g., trace gas concentrations) following the protocol of the Atmosphere Model Intercomparison Project (AMIP, Gates (1992)). In this
configuration, the atmosphere and land models are active while sea surface temperature (SST) and sea-ice cover fraction (SIC) are prescribed. The so-called "data ocean" model reads, interpolates in time and space, and passes the input SST to the CESM coupler, which calculates fluxes between the atmosphere and ocean (CESM Software Engineering Group, 2014). The Community Ice Code version 4 (CICE4) is run as a partially prognostic model by reading prescribed sea ice coverage and atmospheric forcing from the coupler to calculate ice-ocean and ice-atmosphere fluxes (Hunke and Lipscomb, 2010). The ocean and sea





245 ice components communicate with the atmosphere model once per day. The land component is the Community Land Model
version 4 (CLM4; Lawrence et al. (2011)), which simulates vertical exchanges of energy, water, and tracers from the subsurface
soil to the atmospheric surface layer. CLM4 takes a hierarchy-tiling approach to represent unresolved surface heterogeneities,
distinguishing physical characteristics among different surface land covers (e.g., vegetated, wetland, lake, urban), soil texture,
and vegetation types (Oleson et al., 2010). While CLM4 is able to simulate the carbon and nitrogen cycles and transient land
250 cover types, these biogeochemical functionalities are turned off. Instead, our simulations use a prescribed vegetation state (leaf
area index, stem area index, fractional cover, and vegetation height) that roughly represents the conditions around year 2000
based on remotely sensed products (Lawrence et al., 2011). The land cover types are also prescribed to roughly represent the
conditions around the year 2000, and fixed throughout the simulations in both the eval and rcp85 experiments. These land
surface settings are again consistent with the models that participated in NA-CORDEX. Note that the spatial resolution of the
255 original data to derive CLM's land surface characteristics varies from 1 km to 1.0°, with 0.5° being considered as the base
resolution (Oleson et al., 2010). These input data are available from the CESM data repository (CESM Software Engineering
Group, 2014). The CLM4 land model, data ocean, and prescribed sea-ice model are also configured to run on the MPAS hori-
zontal grid. This way, the state and flux data between different model components do not need to be horizontally interpolated
during the model integration.

260 **3.2 Input data**

All the input data required to reproduce our simulations are publicly available (see section 7, Code and data availability). The
SST and SIC for the eval run are taken from the ERA-Interim reanalysis (Dee et al., 2011). The 6-hourly ERA-Interim SST and
SIC data are averaged to daily values and provided to the model as input, then bilinearly interpolated to the MPAS grids by the
CESM coupler during model integration. Other model input data include surface topography, initial conditions, and remapping
265 weights between different input data and model grids (Appendix A). All the surface-related input data are remapped to each
MPAS grid prior to the simulations following the CESM1.2 and CLM4 user guide (CESM Software Engineering Group, 2014;
Kluzek, 2010). A set of high-level scripts is now available to help prepare input data for the FAMIPC5 and other similar CESM
experiments (Zarzycki, 2018). Topography input is generated by the stand-alone MPAS-Atmosphere code (init_atmosphere,
Duda et al. (2015)), which uses the GTOPO global 30s topography data (Gesch and Larson, 1996) as the input. The subgrid
270 topography information required by the gravity wave drag and turbulent mountain stress parameterizations are produced by the
NCAR-Topo tool (Lauritzen et al., 2015).

  As stated above, the future simulation is conducted using the pseudo-global warming approach (e.g. Haarsma et al., 2016)
based on the climate-change signal simulated by the MPI-ESM-LR model from the CMIP5 archive. Specifically, annual cycles
of the daily climatolgoical SST and SIC are obtained from the historical and RCP8.5 simulations of the MPI model (ensemble
275 member id r1i1p1), and differences between the two periods are calculated for each day of the year and each grid point. This
daily climatological difference ($\triangle$SST and $\triangle$SIC) is then added to the SST and SIC from the ERA-Interim data, and prescribed
to the model. The annual average $\triangle$SST and $\triangle$SIC are shown in Figure 4e, f.



While the SST and SIC distributions in the present-day period are reasonably simulated by MPI-ESM-LR, regional biases exist over the Southern Ocean, North Atlantic, and off the west coasts of North and South America and South Africa (Figure 4a–d). Because △SST and △SIC are added onto the climatology from ERA-Interim, the future SST and SIC forcing given to CAM-MPAS is different from those in the MPI model over the biased regions. We will briefly assess how the near-surface climate differ between CAM-MPAS and MPI-ESM-LR in Sect. 5.2.2. Other external forcings of solar irradiance, greenhouse gas, ozone and other tracer gas concentrations are the same as the CESM1.2 RCP8.5 simulation conducted for CMIP5, except for the prescribed aerosol concentrations and land cover characteristics being kept the same as the eval simulation (Mearns et al., 2017).

The atmospheric initial condition for the eval experiment is taken from the ERA-Interim data on 1989-01-01_00 UTC for all simulations except for the VR12-46 simulation that used 2000-01-01_00 UTC data. The land initial condition is taken from the year 2000-01-01 from a 0.5° fully coupled CCSM4 simulation for the historical period (CESM, 2016). The CLM4 land state on the 0.5° grid is remapped to the MPAS grids following Kluzek (2010). Starting from these initial conditions, the model is run for one year to spin up the eval simulations. For the future rcp8.5 experiments, the initial condition for each resolution is taken from the 2011-01-01 state of the corresponding eval simulation, followed by 2 years of spin-up simulations. We found that these spinup lengths are sufficient for the CONUS domain, but not necessarily adequate in the deep soil layer for the global domain, particularly in high latitudes (will be discussed in Sect. 5.2).

## 4 Downscaling Dataset

### 4.1 Post-processing

To facilitate comparison with other regional models in the NA-CORDEX model archive, the model output on MPAS's unstructured mesh is remapped to a standard lat/lon regional grid defined by the NA-CORDEX project (the so-called NAM grid; Figure 3). Variable names and units used in CAM/CESM are converted to those of Climate and Forecast (CF) metadata conventions (version 1.6) that are used by NA-CORDEX. Three-dimensional atmospheric variables defined on the terrain-following model coordinate are vertically interpolated to the NA-CORDEX requested pressure levels (200, 500, and 850 hPa). The following describes how such post-processing was performed.

We mainly used the Earth System Modeling Framework (ESMF) library (Balaji et al., 2018) through the NCAR Command Language (NCL) (UCAR/NCAR/CISL/TDD, 2017a) for regridding MPAS output. The ESMF library provides several remapping methods, among which the first-order conserve method is used for extensive variables and fluxes, and the patch recovery method is used for all other variables. For variables required at a specified pressure level, we first linearly interpolate from the model height level to the pressure level, followed by horizontal remapping. The order of the vertical vs. horizontal interpolation is not expected to be important for the accuracy of subsequent analyses (Trenberth, 1995). Note that the three pressure levels available in the post-processed archive are not sufficient to close budget equations of vertically integrated quantities such as moisture and energy (B. Harrop, unpublished result). For moisture budget analyses, data users are encouraged to use the



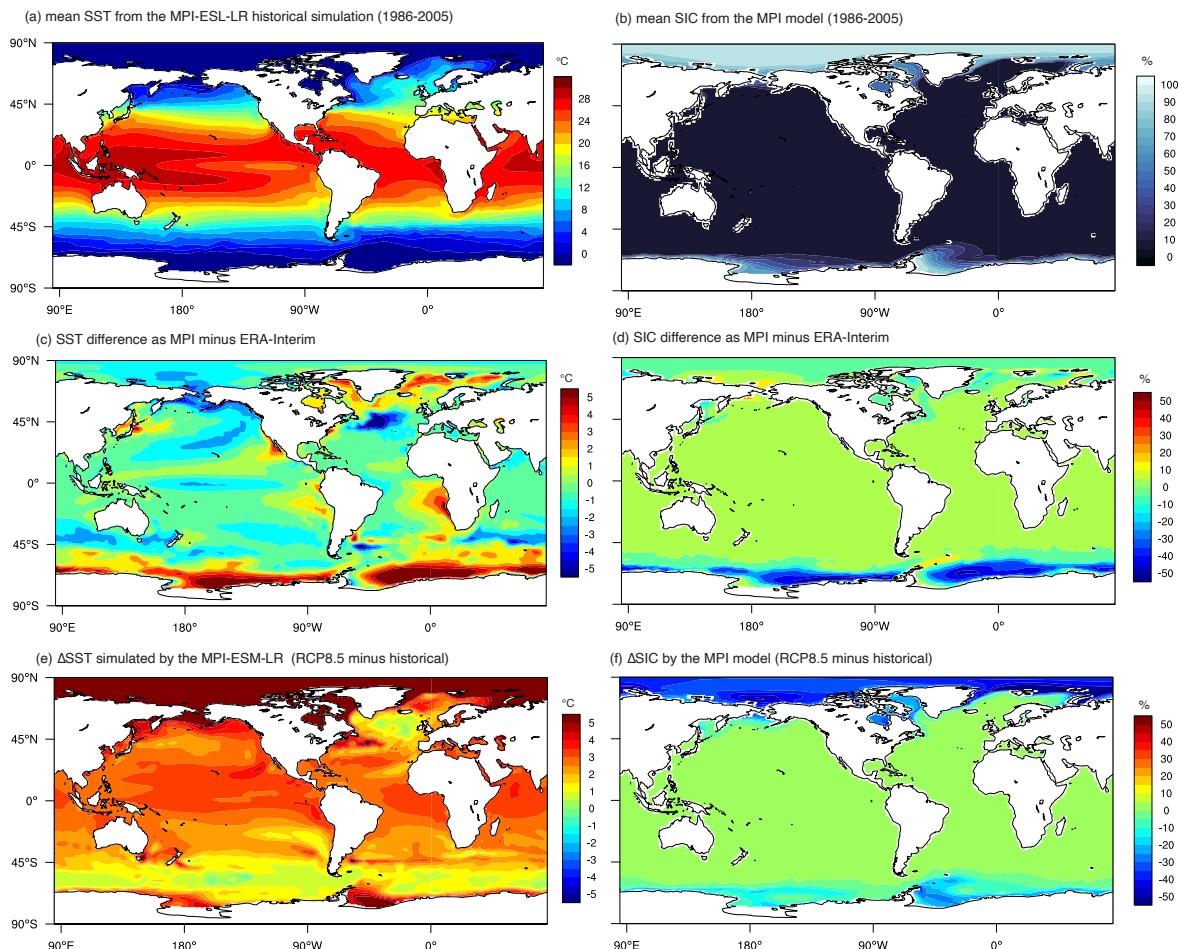

**Figure 4.** Climatological mean sea surface temperature (SST) and sea ice cover fraction (SIC) from the MPI-ESM-LR model: (a) annual-mean SST, (b) annual-mean SIC, (c) SST bias against ERA-Interim , (d) SIC bias, (e) SST change from the historical to RCP8.5 period, and (f) SIC change over the same time periods. The historical and RCP8.5 averages are calculated over the 1986-2005 and 2080-2099 periods, respectively.

vertically integrated moisture fluxes and water vapor path available in the daily variables (Appendix C2). For other variables, it is possible to retrieve them at more pressure levels from the monthly or six-hourly raw model outputs (Appendix B).

    Missing values exist in some variables in the raw model output on the MPAS grids, e.g., soil moisture in the grid points where 100% of its area is covered by ocean/lake/glacier. The locations of such missing values do not change with time, and the corresponding grid points are masked when generating regridding weights. Time-varying missing values arise during

vertical interpolation to a pressure level over the areas where surface topography crosses the target pressure level. We followed the guidance provided by the NCL website to regrid such time-varying missing values (UCAR/NCAR/CISL/TDD, 2017b).



Specifically, we first remap a binary field defined on the source MPAS grid in which all values are one where the vertically interpolated pressure-level variable is missing, and zero everywhere else. By remapping such a field from the original MPAS grid to the destination grid, we can identify by non-zero values which destination grid points are affected by the missing values
on the original grid, and the remapped pressure-level variables in these destination grid boxes are set missing. This rather cumbersome procedure can be replaced by a remapping utility recently enhanced in the NetCDF Operator (NCO) (Zender, 2017).

**Table 4.** Mean (mm d$^{-1}$), variance (mm$^2$ d$^{-2}$), kurtosis, and selected percentiles (mm d$^{-1}$) of daily precipitation sampled from the central-eastern United States (30-47 °N, 85-105°W) on the original and remapped grids with grid spacings similar to the original. The first-order conserve remapping method is used. The analysis domain covered in the WRF output on the curvilinear grid is slightly smaller than the domain used for MPAS, hence the disagreement in statistics between these two model groups.

| Model, grid | mean | variance | kurtosis | 95th | 99th | 99.9th | 99.99th |
|---|---|---|---|---|---|---|---|
| VR50-200 original grid | 2.02 | 30.35 | 65.71 | 10.19 | 26.87 | 60.17 | 97.30 |
| VR50-200 remapped to NAM-44i | 2.00 | 28.04 | 57.34 | 10.08 | 26.17 | 56.43 | 89.37 |
| VR25-100 original grid | 2.10 | 32.37 | 57.00 | 10.81 | 28.27 | 60.09 | 95.70 |
| VR25-100 remapped to NAM-22i | 2.09 | 29.85 | 53.17 | 10.74 | 27.78 | 58.45 | 92.20 |
| VR25-100 NAM-22i, 1991-1995 | 2.03 | 28.75 | 75.26 | 10.43 | 26.50 | 58.75 | 97.51 |
| VR25-100 NAM-22i, 1996-2000 | 2.12 | 31.39 | 55.35 | 10.83 | 28.77 | 60.73 | 95.51 |
| VR25-100 NAM-22i to original | 2.10 | 29.40 | 51.16 | 10.78 | 27.70 | 57.84 | 90.47 |
| VR12-46 original grid | 2.19 | 35.09 | 65.92 | 11.26 | 28.59 | 63.60 | 106.43 |
| VR12-46 remapped to NAM-11i | 2.18 | 34.38 | 64.72 | 11.20 | 28.30 | 62.76 | 104.75 |
| UR120 original grid | 2.06 | 34.10 | 74.48 | 9.91 | 28.91 | 65.73 | 103.57 |
| WRF 25km original grid | 2.81 | 52.05 | 49.60 | 15.67 | 35.12 | 69.07 | 116.62 |
| WRF 25km remapped to NAM-22i | 2.81 | 48.74 | 45.07 | 15.32 | 34.15 | 66.24 | 109.88 |
| WRF 25km NAM-22i to original | 2.81 | 48.64 | 42.89 | 15.41 | 34.08 | 65.81 | 108.32 |

NA-CORDEX documents minor artifacts due to interpolation by the patch recovery method (e.g., small negative values for non-negative variables such as relative humidity) (Mearns et al., 2017). The influence of horizontal regridding, or interpolation,
on the statistics has also been noted by previous studies (Chen and Knutson, 2008; Diaconescu et al., 2015). To understand the effect of regridding in our post-processing, Table 4 compares selected statistics calculated on the original and remapped daily precipitation using different remapping methods: bilinear, patch recovery, first-order conserve, and second-order conserve available from the ESMF library (Balaji et al., 2018). The regridding effect on the (spatial) mean is negligibly small using any of the regridding methods. As shown in Fig. 5a, the global annual mean precipitation (3.004 mm d$^{-1}$) is nearly identical (to
the accuracy of 10$^{-3}$ mm d$^{-1}$) for the original and the regular latitude-longitude 1° grid after remapping.





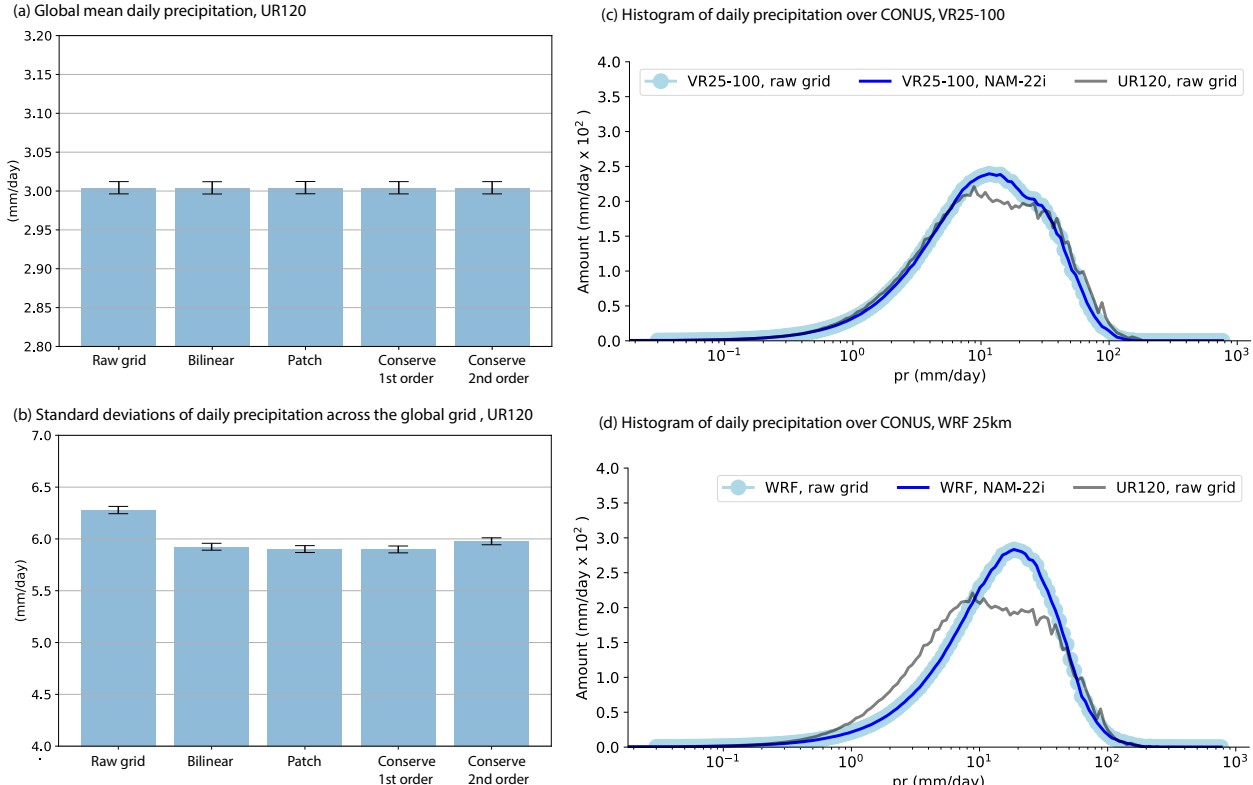

**Figure 5.** Comparison of regridded daily precipitation using four different methods: (a) global annual average of precipitation in UR120 and (b) standard deviations of precipitation across the global grid in UR120, (c) daily rain rate amount distributions over CONUS calculated on the original MPAS grid in VR25-100 (dark blue line), on the remapped (1st order conserve) latitude-longitude NAM22 grid in VR25-100 (light blue circles), and on the original MPAS grid in UR120 (gray line), and (d) same as (c) but calculated on the original WRF grid in the NA-CORDEX WRF-25km simulation (dark blue), on the remapped NAM22 grid in the WRF-25km simulation, and the UR120 histogram as in (c) for comparison. The statistics are based on a 10-year period from 1990 to 1999, and the error bars in (a) and (b) shows 95 % confidence interval based on the year-to-year variance. The distributions of rain rate amount is calculated following Pendergrass and Hartmann (2014), using the minimum rain rate of 0.029 mm/day and a 7 % spacing.

The variance loss due to remapping is typically ≈ 6-8% for daily precipitation. The magnitude of variance loss depends on which variable is remapped — a variable with a smoother spatial structure than precipitation (e.g., atmospheric temperature) is less affected by regridding. At the global scale, ≈ 6-8% loss of variance can be larger than year-to-year sampling variability, as illustrated in Fig. 5b. The second-order conserve method retains the spatial variance slightly better than the other methods. At regional scales, sampling uncertainty from different time periods (here each sample is five years long) can be as large as the smoothing effect. This is illustrated in Table 4 (from the third to sixth rows) based on the statistics of daily precipitation in the CONUS sub-domain east of the Rockies, calculated on the original VR25-100 grid and conservatively remapped to the



NAM-22i grid. We avoid the Rockies and other mountainous regions where year-to-year variability is so large that our sample
size is not long enough to reliably estimate spatial variances. The third and fourth rows present the statistics on these two grids
from the years 2001-2005, while the fifth and sixth rows are from the 1991-1995 and 1996-2000 time periods, respectively,
on the NAM-22i grid. The five-year average of spatial variance is 32.37 mm d$^{-1}$ on the original grid for 2001-2005, which is
reduced to 29.85 mm d$^{-1}$ after regridding. The spatial variance from the other 5-year period can differ from the variance from
2001-2005 by as much as the regridding loss. Similar magnitudes of smoothing effect and sampling uncertainty are also found
in kurtosis and extreme values represented by the 95th to 99.99th percentiles. These differences are not visible on the daily
precipitation histograms, calculated on the original VR25-100 grid, the remapped NAM-22i grid, and the UR120 output on its
raw MPAS grid, for the same CONUS sub-domain (Fig. 5c). The two histograms of VR25-100 are visually identical, and the
difference from the UR120 precipitation is clearly distinguishable.

Two other points notable in Table 4 are: 1) the smoothing effect becomes weaker with finer grid resolutions based on the three
VR resolutions, and 2) successive remapping back from the regional NAM-22i to the original grid (the seventh row) leads to a
further loss of the variance and other moments but to a lesser degree compared to the first remapping. Similar smoothing effects
from the first and second remapping is observed in the outputs from the WRF model on a 25-km grid from the NA-CORDEX
archive. Despite the fact that WRF uses a regular latitude-longitude grid that is similar to the NAM-22i grid, regridding effects
on the selected statistics resemble those on the CAM-MPAS outputs. For example, regridding VR25-100 outputs loses ≈ 8 %
of the daily precipitation variance by the first remapping, while the 25 km WRF simulation loses 6 %. The histograms of daily
precipitation in the WRF 25km simulation are shown in Fig. 5d, again confirming that the histograms are not visually affected
by regridding. Given such a priori knowledge of the regridding effect and sampling uncertainty at regional scales, we do not
expect that the remapping effect would seriously affect statistical inference of regional climate metrics.

## 4.2   Data Repositories

Post-processed monthly and daily variables in the "essential" and "high priority" list of the NA-CORDEX archive (Mearns
et al., 2017) are accessible from the Pacific Northwest National Laboratory DataHub, and other variables and temporal fre-
quencies are available from the NERSC High Performance Storage System (HPSS), made accessible through web browsers by
the NERSC Science Gateway Service (see the Code and data availability section). All variables requested from the experiment
protocol are two-dimensional at a single level. Appendix C lists the post-processed variables.

File names, attributes, and coordinates of the reported variables and their file specification follow the CORDEX archive
design (Christensen et al., 2014) and NA-CORDEX data description (Mearns et al., 2017). The file name is composed of the
following elements:
[variable name].[scenario].[driver].[model name].[frequency].[grid].[bias correction].[start month]-[end month].[version].nc.
In the CAM-MPAS dataset, the scenario is either "eval" for the historical period or "rcp85" for the pseudo-warming future
simulation. The driver is "ERA-Int" for the historical period and "ERA-Int-MPI-ESM-LR" for the rcp85 case. Post-processing
of the current CAM-MPAS simulations does not involve any bias corrections, hence it is labeled as "raw". The major version
refers to different production simulations and the minor version refers to changes/corrections in post-processing stage. The





publicly available CAM-MPAS outputs are either "v3" or "v3.1"; the major version is three because it was necessary to re-run simulations twice due to major changes in model configurations, and the minor revision involves a different treatment of missing values arising from vertical interpolation to a pressure level (see Sect. 4.1). With the other straightforward file name elements, an example file name for a daily precipitation data in the historical run of CAM-MPAS VR50-200 reads as:

pr.eval.ERA-Int.cam54-mpas4.day.NAM-44i.raw.198901-201012.v3.nc

and in the future pseudo-warming:

pr.rcp85.ERA-Int-MPI-ESM-LR.cam54-mpas4.day.NAM-44i.raw.207901-210012.v3.nc.

Raw CAM-MPAS outputs on the global MPAS grid (i.e., not remapped to a regional latitude-longitude grid) are also available from the NERSC HPSS space. Appendix B provides more information about the MPAS unstructured mesh, links to the archive directory, and other resources to help analyze the raw MPAS data. The NERSC data archive also contains example scripts and variables necessary to process model variables on the MPAS grid (e.g.,latitude and longitude arrays).

## 5 Simulations

### 5.1 Computational aspects

In this section, we discuss some computational aspects of our simulations because one of the motivations to use a global VR framework is its computational advantage compared to a global high-resolution simulation. On the other hand, global VR simulations are expected to be more expensive than limited-area model simulations, if not considering the cost for the host GCM simulations that provide boundary conditions. For example, the VR grids used in this study have 1.1 to 2.6 times more grid columns than the limited-area grids used by the RegCM4 and WRF models in the NA-CORDEX and HyperFACETS archives (Tables 2 and F2). Here we do not compare simulation costs of CAM-MPAS VR configurations against regional models, but focus on how the cost of CAM-MPAS simulations differ between the UR and VR grids and between the lower and higher resolutions.

All of our simulations were run at NERSC. The following result is obtained from the production simulations and not a systematic scaling analysis of the CAM-MPAS code nor NERSC systems. The system configurations (e.g., number of MPI tasks) of our production simulations are not only based on good throughput but also on simulation cost as well as expected queue wait time (Figure D1), which often accounts for the majority of the total production time (e.g., for VR25-100, the average queue wait time is approximately three times the actual computing time). All simulations used only the distributed-memory Message Passing Interface (MPI) parallelism, i.e., shared-memory parallelism (OpenMP) is not used. The main computing system at NERSC switched from Edison to Cori when the production simulations of the CAM-MPAS model were starting (NERSC, 2021). The newer system Cori is partitioned into two sub-systems, Cori-Haswell (HW) and Cori-Knights Landing (KNL). As discussed below, the CESM-CAM-MPAS code showed large differences in performance on KNL and other systems, posing a significant impact on our production cost. Interested readers are referred to Appendix D for further details of our run-time configurations and the characteristics of the NERSC systems.





Three simulations that are not part of the CAM-MPAS downscaling dataset are also included in the following as references:

1) the default FV dynamical core on the nominal 1° grid ("FV 1°"), 2) the same model configuration as UR120 but using the newer version of the CAM-MPAS model that will be released as an official option of CESM2 ("UR120-new"), and 3) CAM-MPAS on a quasi-uniform 30km grid ("UR30"). These three simulations were run for other projects, but with a similar set of file outputs (monthly, daily, 6-hourly, 3-hourly, and hourly outputs) for more than five years. All simulations use the same CAM5.4 physics with prescribed aerosol.

**Table 5.** Simulation throughput and cost. The simulation cost is based on so-called "NERSC hour" (= number of nodes $\times$ number of hours $\times$ machine-dependent charge factor $\times$ queue priority factor), assuming the "regular" queue, and shown in the units of $10^3$ NERSC hours per simulated years (NERSC hr sim.yr$^{-1}$). Throughput (sim.yr day$^{-1}$) is an average of at least 60 jobs with the standard deviations shown in parentheses. $N_{calc}$ is the number of time steps per day over all grid boxes $\times 10^{-7}$. Most of the samples are production runs, except for UR120-new, FV 1°, and UR30, which are not the part of the dataset described in this paper but shown as references.

| Model grid | Columns | $N_{calc}$ | System | MPI tasks | Nodes | Col./task | Throughput | Cost |
|---|---|---|---|---|---|---|---|---|
| UR240 | 10242 | 4.7 | Edison | 120 | 5 | 85 | 11.9 (0.26) | 0.6 |
| UR240 | 10242 | 4.7 | KNL | 120 | 2 | 85 | 2.8 (0.12) | 1.4 |
| UR120 | 40962 | 31.5 | Edison | 384 | 16 | 107 | 5.5 (0.20) | 4.5 |
| UR120 | 40962 | 31.5 | KNL | 640 | 10 | 64 | 1.9 (0.08) | 10.1 |
| UR120-new | 40962 | 31.5 | KNL | 640 | 10 | 64 | 3.5 (0.27) | 5.5 |
| FV 1° | 55296 | 25.5 | KNL | 640 | 10 | 86 | 2.1 (0.06) | 9.1 |
| VR50-200 | 34306 | 73.8 | Edison | 240 | 10 | 143 | 2.3 (0.16) | 6.7 |
| VR50-200 | 34306 | 73.8 | HW | 256 | 8 | 134 | 2.3 (0.09) | 11.7 |
| VR50-200 | 34306 | 73.8 | KNL | 1024 | 16 | 34 | 1.6 (0.06) | 19.4 |
| VR25-100 | 137218 | 509.6 | Edison | 960 | 40 | 143 | 1.4 (0.10) | 43.9 |
| VR25-100 | 137218 | 509.6 | KNL | 2560 | 40 | 54 | 0.7 (0.05) | 109.7 |
| VR12-46 | 655362 | 3623.9 | Edison | 4320 | 180 | 152 | 0.8 (0.02) | 345.6 |
| VR12-46 | 655362 | 3623.9 | KNL | 5120 | 80 | 128 | 0.2 (0.02) | 713.0 |
| VR12-46 | 655362 | 3623.9 | KNL | 6144 | 96 | 107 | 0.3 (0.02) | 697.3 |
| UR30 | 655362 | 1409.3 | KNL | 6400 | 100 | 102 | 0.4 (0.02) | 442.1 |

Figure 6a visualizes the simulation costs vs. total MPI tasks used, as often used in cost scaling studies. Table 5 lists the numerical values used in the figure. Although scatters in the data from different computing systems are notable, there is a clear trend to which we can fit a curve. The blue line represents a power function ($y = ax^b$) fitted to the simulation costs in the log-log space. The exponent $b$ (= the slope of a straight line on the log-log plot) is 1.54 with a 95% confidence interval of 0.50, exhibiting a weak but non-linear increase. The non-linear increase is expected because linearly increasing cost is only possible

for an idealized case, also shown in the figure. The green line represents an ideal situation that the parallel part of the code speeds up linearly with additional resources (an ideal weak scaling, eqn. 5.14 in Hager and Wellein (2011)), whose cost thus



increases linearly with the number of MPI ranks (slope of 1). The orange line of a constant cost applies only to the case where the size of the problem (e.g., number of grid columns) stays the same so that using more resources shortens the simulation time. This is an ideal "strong scaling" and not applicable to the cost scaling for different resolutions over a fixed global domain.

It is obvious from this comparison that larger resource use for higher resolutions on a fixed domain size, such as the global domain, *always* increases the computing cost non-linearly.

There are several reasons for the non-linear increase of the simulation cost against resources used, such as communication and load imbalance (Hager and Wellein, 2011; Heinzeller et al., 2016). For estimating the simulation cost of a given MPAS grid, we found that it is simpler to use the number of calculations (physics and dynamics timesteps) per simulated day across all the

grid boxes in the global domain; $N_{calc}$ = (number of grid columns) × (number of vertical levels) × (number of timesteps per day). Plotting simulation costs as a function of $N_{calc}$ (Figure 6b), the fitted curve exhibits a slope of approximately one. Looking at $N_{calc}$ as a function of the number of grid columns, it appears to be separated into two groups of VRs and URs, indicating the timestep constraint from the high-resolution domains in VRs (Figure 6c). The least-square fitted power functions have the exponents of 1.45 for both VR meshes and UR meshes. This weak non-linearity presumably comes from the dependence of

timestep length on grid spacing, which then becomes an additional implicit dependence on the numbers of grid columns.

As a specific example of VR vs. UR comparison, we take VR25-100, UR30, and UR120 because the latter two URs have comparable grid spacings to the high- and low-resolution regions of the VR25-100 grid. We use $N_{calc}$ of the simulations conducted on KNL to gauge the computational advantage of the VR25-100 against UR30, a uniform high-resolution simulation, as well as the extra cost added by the regional refinement to a uniform low-resolution simulation, UR120. The actual values of

$N_{calc}$ for these three resolutions are shown in the third column of Table. 5, which suggest UR30 to be 48 times more expensive than UR120, while VR25-100 being 16 times more costly than UR120. The actual simulation cost closely follows the $N_{calc}$ scaling; one simulation year of UR30 (480.0 ×$10^3$ NERSC $hr\ sim.yr^{-1}$ ) is 48 times more expensive than that of UR120 (10.1 ×$10^3$ NERSC $hr\ sim.yr^{-1}$.) The actual cost of VR25-100 is just 11 times that of UR120, lower than those expected from $N_{calc}$, possibly reflecting the error from using an empirical curve fitted to three different systems in the single KNL

system. In this case, VR25-100 achieves a factor of four computational advantage compared to UR30 for obtaining a similarly high-resolution grid over CONUS.

A couple of other points are noted in Table. 5 and Fig. 6. First, the computational costs of CAM-MPAS UR120 and the default dynamical core FV 1° are comparable (1.9 vs. 2.1 sim.yr. day$^{-1}$ for CAM-MPAS UR120 and CAM-FV 1°, respectively). Second, the model throughput (cost) of VR12-46 is 0.2 sim.yr. day$^{-1}$, half (double) that of UR30, despite the fact that these

two grids have the same number of columns and the simulations are run with similar numbers of columns per MPI task. The main reason for the difference is likely the shorter timesteps (about 1/3) in VR12-46 than in UR30 due to the numerical constraint imposed by the smallest grid spacing in the high-resolution domain. Lastly, we get consistently lower throughput and higher costs on Cori-KNL than on the other two systems. Our experiment and previous studies (Barnes et al., 2017; Dennis et al., 2019) suggest a few compounding reasons (Appendix D): inefficient memory management for some global arrays,

poor vectorization, and less focus on shared-memory parallelism of the CAM5/MPASv4 source code, which are not aligned well with the wider-vector and many-core architecture of KNL. However, the shorter expected queue time on KNL than HW





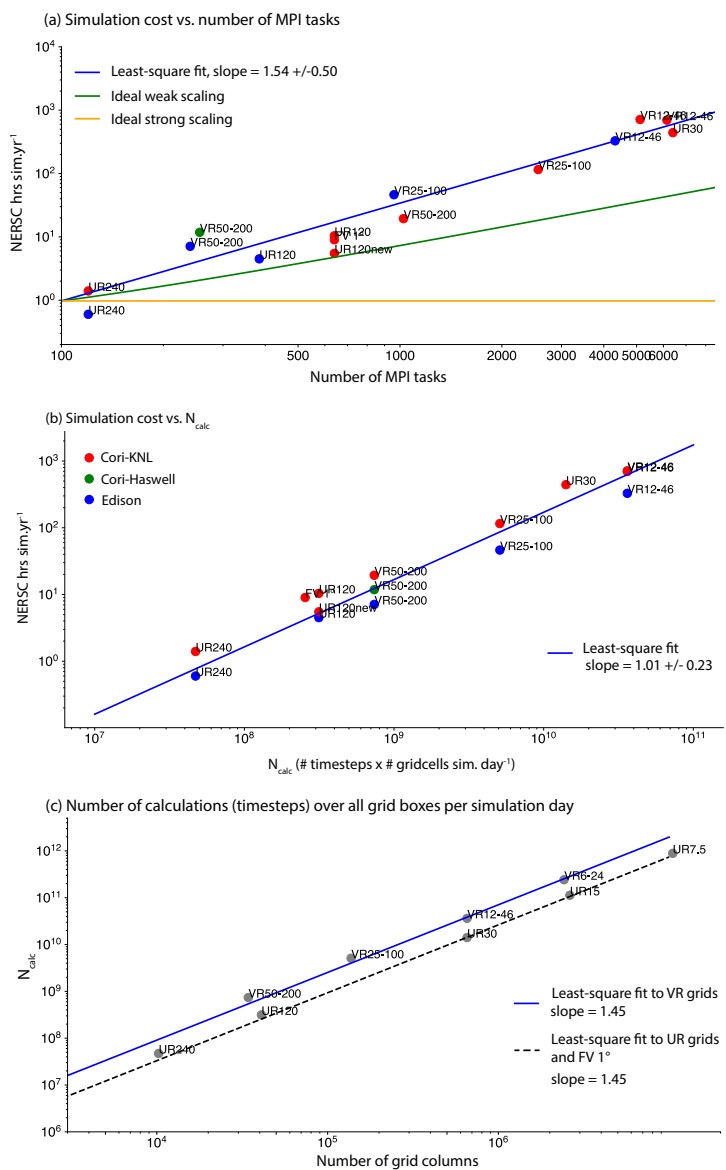

**Figure 6.** Graphs showing the relationship between (a) simulation cost in terms of NERSC hours per simulated years (NERSC $hr\ sim.yr^{-1}$) and number of MPI tasks, (b) simulation cost and $N_{calc}$, (number of calculations = physics and dynamics timesteps per simulated day across the global domain), and (c) $N_{calc}$ and number of grid columns. The parameters of the fitted linear lines (blue curves, linear in the log space), $y = a + bx$, are shown in the legend. UR120-new refers to the UR120 simulation using the new CAM-MPAS code under development. In (c), we added data points for a variable-resolution 6-24km mesh (VR6-24) as well as uniform resolution with 15 and 7.5 km gridcells (UR15 and UR7.5) by using their numbers of grid columns and scaling the model timestep as described in Sect. 3.




(Figure D1) makes KNL our main system for production. The weaker performance of the experimental CAM-MPAS code on KNL leads to a higher computational cost than our initial estimate for VR12-46, limiting the length of VR12-46 simulations to be half of other simulations. More importantly, the code characteristics described above are not necessarily unique to the CAM5/MPASv4 codes but may be common in other global or regional climate models in which many lines of the codes are written by domain scientists with little attention to code optimizations. Such climate models are not likely to be efficient on emerging, more energy-efficient HPC architecture similar to KNL for having wider vector units and more cores per node (and less memory per core) than previous systems. For example, two new systems being deployed to HPC centers in the United States –Perlmutter to NERSC (NERSC, 2022) and Derecho to the NCAR-Wyoming Supercomputing Center (NCAR Research Computing, 2022)– share such characteristics in their CPU nodes.

Fortunately, some of the computational problems with the CAM-MPAS model have been resolved through the MPAS-Atmosphere optimization, on-going effort to port the later version 6 of MPAS-Atmosphere to CESM2 (the SIMA project), and other numerous changes across the CESM source code from CESM1.5 to CESM2. Those updates lead to almost 80% speed up of the UR120 throughput as can be seen on the UR120 and UR120-new simulations in Table 5. Some of the speed-up comes from different compiler optimizations used for the two simulations, but the code development plays a major role in this performance improvement. The Cori system is retiring but the computational advantage of the new code is expected to be applicable to other systems including the new NERSC system Perlmutter. We expect decadal simulations on the VR12-46 grid or even convection-permitting VR meshes will be feasible using the newer CAM-MPAS code, or SIMA atmospheric general circulation model with MPAS as its dynamical core option. Multi-season convection-permitting simulations have been already carried out with the new SIMA-MPAS model (Huang et al., 2022).

## 5.2 General Characteristics of Simulated Climate

We briefly review selected aspects of the simulated climate. The focus here is the climate statistics at the global-scale and over the regions outside the VR high-resolution domain of North America. This is because, although the post-processed datasets cover a broad area encompassing the NA-CORDEX domain (Figure 3), the limited area grid does not allow one to infer remote sources of large-scale forcings and their dependency on model resolution, which may be important to understand processes responsible for projected changes within the high-resolution domain. Appendix E presents additional figures and a table. For the downscaled regional climate, Appendix F provides a general overview of the model performance focusing on the COUNS region. The main findings of the regional assessment is that the performance metrics of precipitation improves with higher resolution, but the results are more mixed for other variables. Also, resolution-sensitivity of precipitation becomes weaker within the North American domain compared to the global statistics, which is shown below. A separate, systematic investigation of the regional climate in comparison with other limited area models is being conducted (Sakaguchi et al., 2021) and will be reported elsewhere.



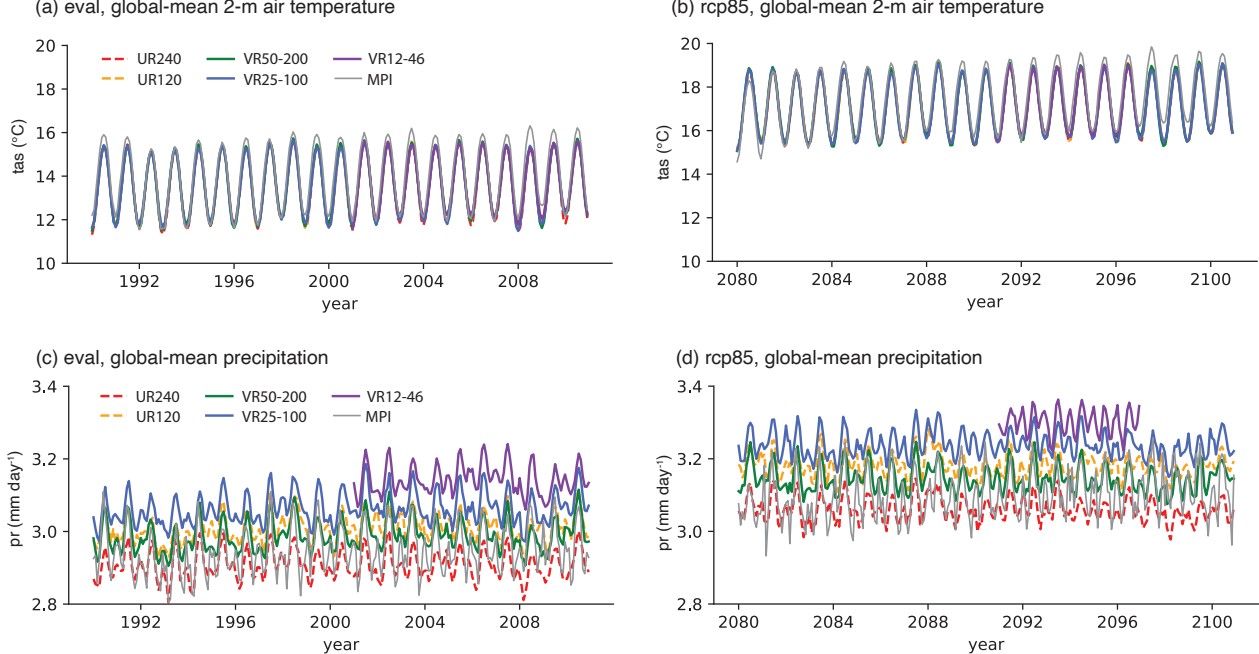

**Figure 7.** Time series of monthly, global mean (a) near-surface air temperature (TAS) in the present-day (eval) simulations, (b) TAS in the future (rcp85) simulations, (c) precipitation (PR) in the present-day (eval) simulations, and (d) PR in the future (rcp85) simulations. The shorter VR12-46 simulation appears only in the last 11 years.

### 5.2.1 Present-day Climate

The time evolution of global mean near-surface air temperature (TAS) is nearly identical across the resolutions (Figure 7a), in-
dicating a strong constraint by the prescribed SST. In contrast, global mean precipitation exhibits systematic differences among
the resolutions such that it monotonically increases with finer resolution; UR240 simulates the lowest global mean precipita-
tion, followed by VR50-200, UR120, VR25-100, and VR12-46 (Table 6, Figure 7c), indicating that in the VR simulations the
coarse-resolution domain dictate the resolution sensitivity at the global scale. We see in Figure 7c that the global precipitation
of MPI-ESM-LR is similar to those of UR240 and VR50-200, the two resolutions closest to the MPI model resolution.

Table 6 indicates that this monotonic increase is mainly contributed by convective precipitation, rather than large-scale
precipitation. The trend of increasing convective precipitation with higher resolution is opposite to what previous studies found
about the lineages of CAM-physics (Williamson, 2008; Rauscher et al., 2013; Wehner et al., 2014; Herrington and Reed,
2020), presumably because of our tuning of the ZM convection scheme by the convective time scale (Sect. 2.2). Other notable
resolution sensitivities are reductions of cloud fraction and vertically integrated cloud liquid and ice mass concentrations,
which then bring about resolution-sensitivities to cloud radiative forcing and radiative fluxes. Reduction of cloud amount with
higher resolutions is noted by previous studies (Pope and Stratton, 2002; Williamson, 2008; Rauscher et al., 2013; Herrington



and Reed, 2020). For example, Pope and Stratton (2002) found a reduction of the global-mean cloud liquid-water path by 12 $gm^{-2}$ by refining grid spacing from $\approx$ 280 km to 90 km in the HadAM3 model. Herrington and Reed (2020) attributed reduced cloud amount to stronger subsidence outside convective regions, which is linked to more intense resolved upward 500 motion within the convective regions at higher resolution. We speculate the same processes operate in our simulations with additional complexities due to our tuning of the ZM convection scheme.

**Table 6.** Global and annual means of selected variables from present-day (eval) simulations, taken from the AMWG diagnostic package (Atmospheric Model Working Group, 2014). Abbreviations in variable names are: top-of-atmosphere (TOA), short wave radiative flux (SW), long wave radiative flux (LW), short wave cloud radiative forcing (SWCF), and long wave cloud radiative forcing (LWCF). Observational and reanalysis data (Obs) are provided through the AMWG diagnostic package and listed in Table E1. Averages are shown for variables for which multiple observational data are available.

| Variable | UR240 | UR120 | VR50-200 | VR25-100 | VR12-46 | cam5.4 1deg | Obs |
|---|---|---|---|---|---|---|---|
| sfc. air temperature (K) | 287.08 | 287.14 | 287.17 | 287.12 | 287.28 | – | 287.58 |
| precipitation (mm d$^{-1}$) | 2.91 | 3.01 | 2.99 | 3.06 | 3.14 | 2.96 | 2.68 |
| convective precip. (mm d$^{-1}$) | 1.81 | 1.83 | 1.89 | 1.93 | 2.00 | - | – |
| large-scale precip. (mm d$^{-1}$) | 1.10 | 1.18 | 1.10 | 1.13 | 1.15 | - | – |
| precipitable water (kg m$^{-2}$) | 26.12 | 25.82 | 25.81 | 25.56 | 25.35 | 25.77 | 24.70 |
| column cloud liquid (g m$^{-2}$) | 54.17 | 52.92 | 53.93 | 53.64 | 39.83 | – | – |
| column cloud ice (g m$^{-2}$) | 22.23 | 22.26 | 19.31 | 17.52 | 14.79 | – | – |
| total cloud fraction (fraction) | 0.64 | 0.62 | 0.64 | 0.63 | 0.59 | 0.66 | 0.67 |
| TOA SWCF (W m$^{-2}$) | -50.31 | -49.06 | -49.48 | -48.79 | -42.89 | -51.00 | -49.96 |
| TOA LWCF (W m$^{-2}$) | 26.21 | 25.16 | 25.13 | 23.88 | 21.46 | 25.41 | 27.87 |
| TOA LW out (W m$^{-2}$) | 233.82 | 236.55 | 236.94 | 239.52 | 243.39 | 234.22 | 237.53 |
| TOA SW net (W m$^{-2}$) | 238.40 | 239.66 | 239.34 | 239.99 | 246.65 | 237.51 | 239.72 |
| Max zonal mean UA200 (m s$^{-1}$) | 34.7 | 33.6 | 34.2 | 32.9 | 31.9 | 35.4 | 31.4 |

Figure 8 examines the spatial patterns of TAS and precipitation biases of VR25-100. We show VR25-100 as an example because the bias patterns are generally similar at the other resolutions (Figures E1, E2). As with CAM5.4 and other climate models (Morcrette et al., 2018), the simulated TAS is too warm over the mid-latitude continents including the central United 505 States (Figure 8a). Little difference from ERA-Interim is seen over the ocean, but notable exceptions exist over the Southern Hemisphere storm track ($\approx$ 0.5 °C) and the Arctic ($> |4|$ °C). The TAS bias appears similar to that of CAM5.4 with the default 1° FV dynamical core (Atmosphere Model Working Group, 2015), indicating a more important role of physics parameterizations than resolution or dynamical core for the bias (Appendix E).

The resolution-sensitivity of the global mean precipitation (Table 6c) originates mostly from the tropics between 20°S and 510 20°N (Figures 8b, 9a) where the model overestimates precipitation compared to GPCP. This regional bias generally becomes worse with higher resolution. While the tropics is far away from the downscale target of North America, tropical precipitation





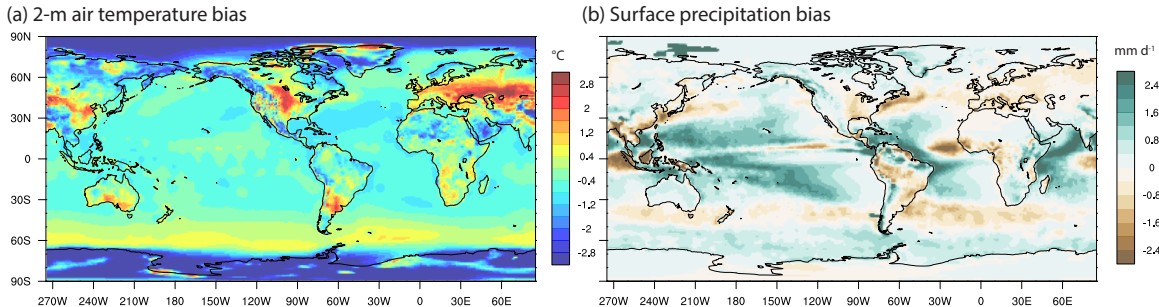

**Figure 8.** Difference of climatological mean (a) 2-m air temperature over the 1990-2010 period between CAM-MPAS VR25-100 and ERA-Interim, and (b) surface precipitation over the 1997-2010 period between VR25-100 and GPCP. The ERA-Interim sea surface temperature and sea ice cover are used as input for the CAM-MPAS AMIP simulations. The CAM-MPAS outputs and the reference data (ERA-Interim and GPCP) are remapped from their original grids to a global latitude-longitude grid with $\approx 0.7°$ grid spacing, a similar resolution to the ERA-Interim grid.

bias may have remote effects on large-scale circulations over the mid-latitudes through Rossby waves and subtropical jets (Lee and Kim, 2003; Christenson et al., 2017; Dong et al., 2018; Wang et al., 2021). Such remote effects seems small over North America but much more prominent in the Southern Hemisphere, consistent with the previous VR CAM-MPAS study

(Sakaguchi et al., 2015). For example, steady changes across resolution appear in the zonal-mean sea-level pressure in the tropics and in the high-latitudes, with clearly greater magnitude in the Southern Hemisphere than in the Northern Hemisphere (Figure 9b). Consistently, zonal-mean zonal wind also shows stronger resolution sensitivities over the tropics and Southern Hemisphere than in the Northern Hemisphere (Figure E3). The CAM Atmosphere Model Working Group (2015) shows a similar sea-level pressure bias in the default CAM5.4, and the apparently large magnitude of the bias depends on which

reanalysis dataset is used as reference. Notably, higher resolution reduces the biases of sea-level pressure and zonal-mean zonal wind over the Southern Hemisphere.

In our global pseudo-warming experiment, differences between the CAM-MPAS and MPI simulations in large-scale circulations are also important to understand the processes underlying regional climate change over North America. Figure 10 compares the climatological mean zonal wind at the 200 hPa level (UA200) and zonal anomalies of 500 hPa geopotential

height (ZG500) from the VR25-100 and MPI model simulations of the historical period. We continue to use VR25-100 as an example because differences between the two models (MPI and CAM-MPAS) are substantially larger than resolution sensitivities of the CAM-MPAS model (not shown). Figure 10a–c indicates that in VR25-100, 1) the mid-latitude (eddy-driven) jet is located at higher latitudes, 2) the subtropical jet over North America is stronger, and 3) the Walker circulations over the Pacific and Atlantic oceans are also stronger than those in the MPI model. Notable differences in ZG500 includes a stronger ridge

in VR25-100 than in the MPI model over the western North America (Figure 10d–f). The stronger ridge and associated static stability, along with different jet locations and strength, indicate that the two models simulate differently the generation and





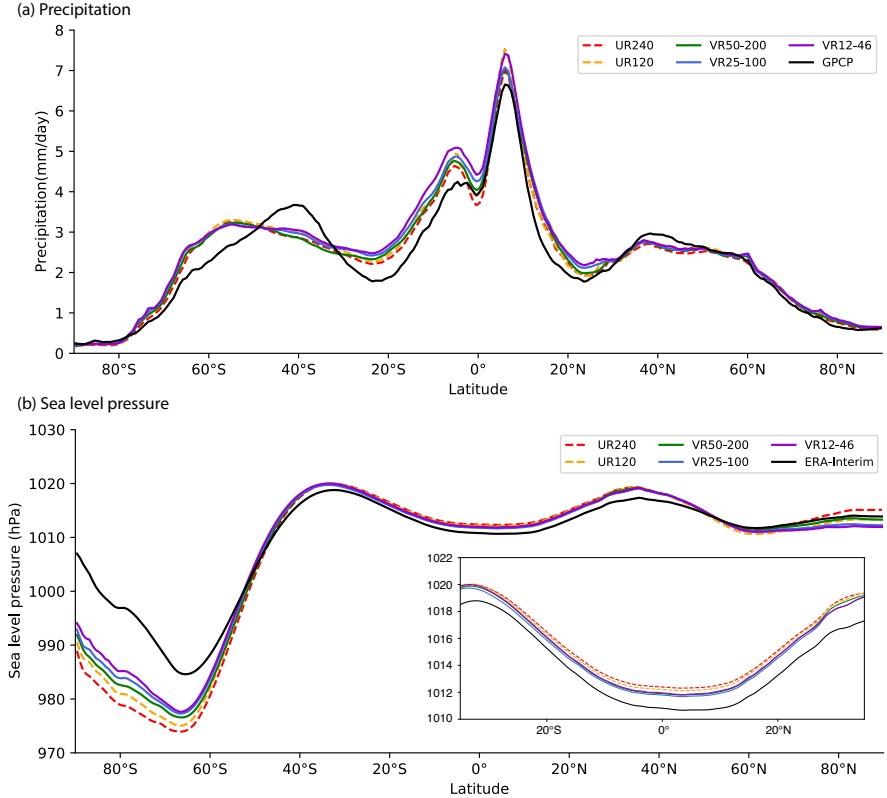

**Figure 9.** Zonal and annual mean (a) precipitation and (b) sea level pressure from the CAM-MPAS simulations and reference data of GPCP in (a) and ERA-Interim in (b). All data are first remapped to $\approx 0.7°$ latitude-longitude grid before taking zonal average. The inset in (b) shows the same mean sea level pressure but only in the region between 35°S and 35°N.

propagation of atmospheric disturbances and local response to them, which are suggested to be important for the hydroclimate of the western and central U.S. (e.g., Leung and Qian, 2009; Song et al., 2021).

### 5.2.2 Future Climate

The global-mean TAS remains insensitive to resolution in the future RCP85 case (Figure 6b). Also similar to the historical period, we see steady increase of global-mean precipitation with finer resolution (Figure 6d). As a result, all the resolutions project similar changes of the global mean precipitation ($\triangle$P) from the historical to RCP85 case within the range of 0.15–0.18 mm day$^{-1}$.

Looking at the spatial patterns, the TAS change ($\triangle$TAS) from the historical to RCP8.5 period in VR25-100 closely follows
the $\triangle$SST patterns derived from the MPI model (by comparing Figure 11a and Figure 4e). The almost identical $\triangle$SST leads to different climatological SST (and TAS) in the two future simulations (Figure 11 b) because $\triangle$SST and $\triangle$SIC from the MPI



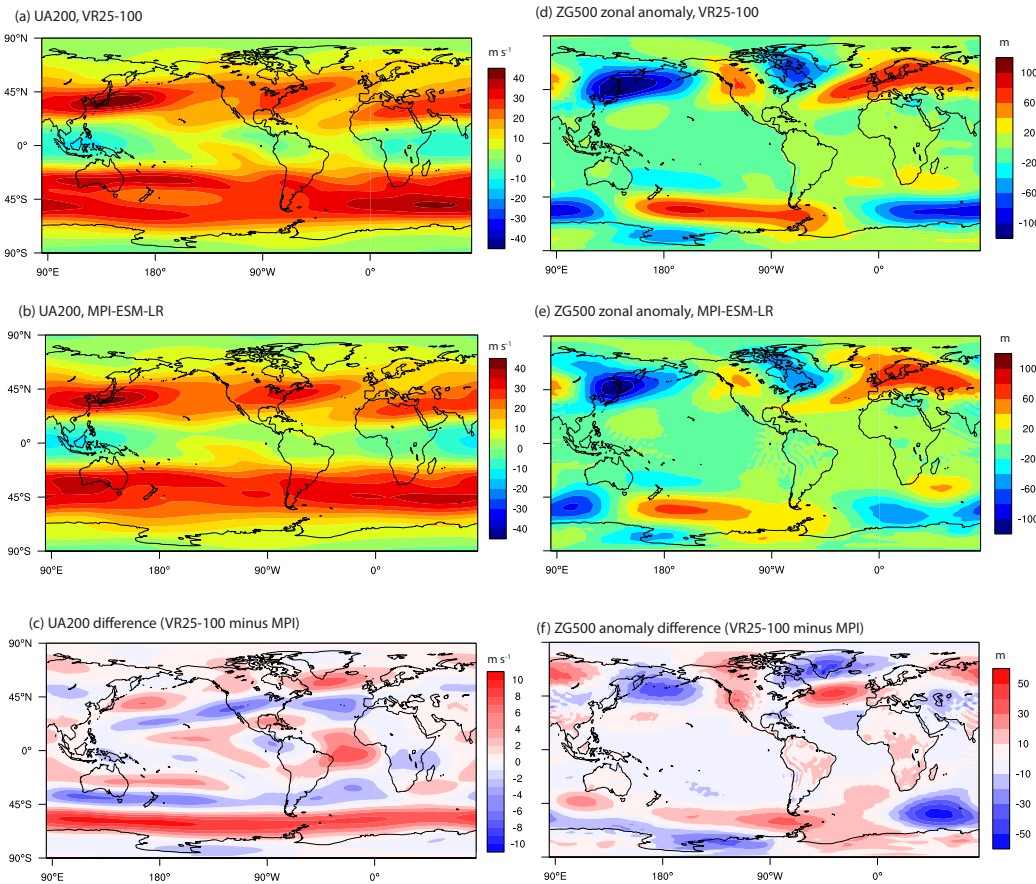

**Figure 10.** Annual mean zonal wind at the 200 hPa level in (a) VR25-100, (b) MPI-ESM-LR, (c) difference between the two simulations, and geopotential height at the 500 hPa level in (d) VR25-100, (e) MPI-ESM-LR, and (f) difference between the two simulations. All data are remapped to ≈ 0.7° latitude-longitude grid by the patch method (Balaji et al., 2018). The wavy patterns in (e) and (f) near the Andes are likely numerical oscillations in the MPI-ESM-LR model (Geil and Zeng, 2015).

model are added to the base state from ERA-Interim instead of the MPI model itself (Figure 11c, d). In the SST difference plot, SST over the Arctic region is substantially warmer in VR25-100 than in the MPI model, while such difference is lacking in SAT (Figure 11b, c). The discrepancy is a result of an assumption in the CESM data ocean model such that SST below -1.8 °C (a typical freezing temperature of sea ice) is reset to this assumed freezing temperature, and the SST shown in the figure is not the input to the model but output from the simulation. In the MPI model without such an assumption, the climatological SST can be as low as -5 °C over the Arctic region. We presume that this SST difference does not directly affect TAS because of the Arctic sea ice cover.






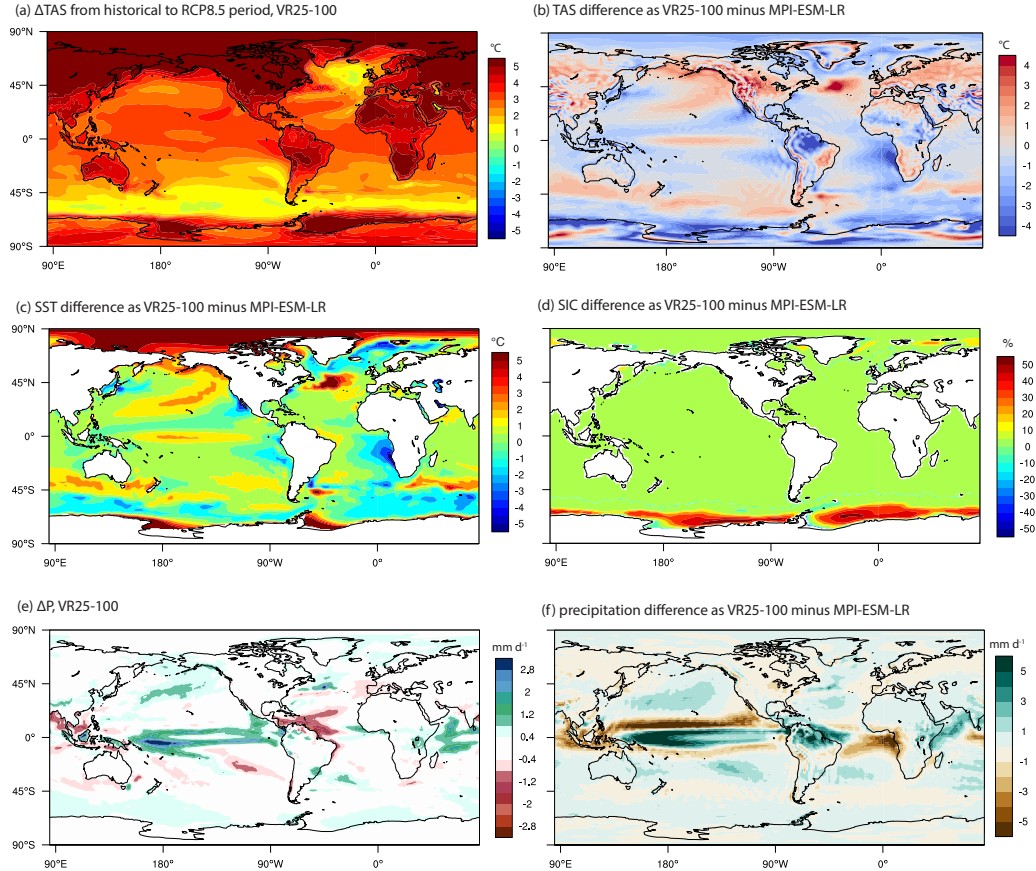

**Figure 11.** Spatial patterns of the near-surface climate change from the historical (1990-2010) to future RCP8.5 case (2080-2100) and the difference of the mean future climate between VR25-100 and the MPI model simulations: (a) simulated change of near-surface air temperature (△TAS) in VR25-100, (b) difference of the mean TAS between VR25-100 and the MPI model in the RCP8.5 simulations, (c) same as (b) but for SST difference, (d) same as (b) but for SIC difference, (e) precipitation change (△P) in VR25-100, and (f) precipitation difference between the two RCP8.5 simulations.

The spatial pattern of △P in VR25-100 is characterized by a marked increase in the tropical Pacific, Arabian Sea, and
Northern Hemisphere storm tracks and by a reduction over the tropical Atlantic Ocean (Figure 11e). These △P responses over the ocean generally agree with the MPI model projection (Figure E4), while the extent of regional features differ, especially in the equatorial region such that the Intertropical Convergence Zones (ITCZ) precipitation is projected to be more intense in a narrower band in VR25-100 than in the MPI model. Over land, △P in the two simulations diverges most notably in the Amazon basin, as well as in Australia, southern Africa, and importantly, North America. These changes over land become more visible
in the ocean-masked contour plots in Figure E5e, f. Those regions are also where we see resolution-sensitivity of △P among the CAM-MPAS simulations (Figure E5b–f), indicating a large uncertainty in the projection of regional hydrological cycles.





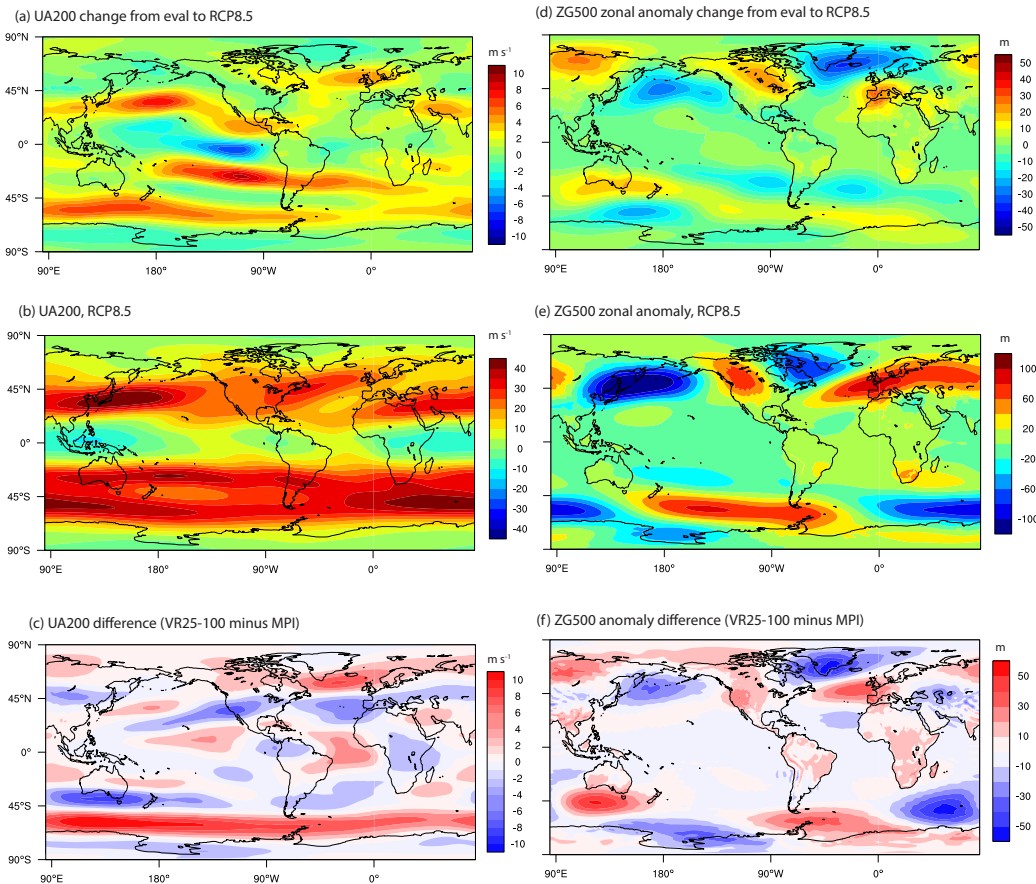

**Figure 12.** Simulated changes of annual-mean upper-level circulations from the historical (1990-2010) to future RCP8.5 case (2080-2100), and the difference of the future climate between VR25-100 and the MPI model simulations: (a) 200-hPa zonal wind change (△UA200) in VR25-100, (b) the future UA200 climatology in VR25-100, (c) UA200 climatology difference between VR25-100 and MPI-ESL-LR in the RCP8.5 period, (d) simulated change of zonal anomaly 500-hPa geopotential height (△ZG500) in VR25-100, (e) the future climatology of ZG500 zonal anomaly in VR25-100, and (f) ZG500 difference between VR25-100 and MPI-ESL-LR.

Turning to the large-scale circulations, the projected change of the 200-hPa level zonal winds (△UA200) in VR25-100 indicate broader and more intense subtropical jets, mid-latitude storm tracks, and Southern Hemisphere polar jet in the end of the twenty-first century (Figure 12a, b). MPI-ESM-LR also projects such changes in terms of the zonal mean circulation (Shaw, 2019), and the spatial patterns of △UA200 are generally consistent between the two models with a pattern correlation of 0.87 (Figure E6). The projected changes of zonal anomaly of 500 hPa geopotential height (△ZG500) in VR25-100 are characterized by the pattern-shift to the east over mid- to high-latitudes in the Northern Hemisphere (Figure 12d, e). The shift






is simulated by MPI-ESM-LR and also found in the CMIP5 multi-model mean response (Wills et al., 2019). Because the responses of these large-scale circulations to the imposed radiative forcings and (identical) ocean warming are similar in the two models, the base-state differences as seen in Figure 10c, f remain nearly unchanged in the future period (Figure 12c, f). Therefore, distinct aspects of the large-scale forcings on the North American climate, as discussed in the previous section, will continue to be seen in the RCP8.5 case.

### 5.2.3 Soil spin-up

Lastly, we would like the readers to be aware of soil spin-up at deep layers in the cold regions outside the refined region. A previous study (Cosgrove et al., 2003) and community experience from the NA-CORDEX (Mearns et al., 2017) suggest that over the CONUS region (i.e., excluding permafrost regions from North America), one year is enough for the model soil state to reach a quasi-equilibrium provided a reasonably realistic soil moisture distribution for the spin-up initial condition (i.e., not an idealized state such as spatially uniform soil moisture content). This is the case for the soil liquid water in the present-day (eval) simulations with one-year spin-up starting from a condition taken from a previous CCSM4 historical simulation (Sect. 3). Using VR50-200 as an example, the CONUS-average soil liquid water does not show a systematic drift at any soil model levels, and neither does the global average (Figure 13a, b, d). The CONUS-average soil ice does not show an obvious trend either (Figure 13c). However, the global-average soil ice in the 10th soil layer shows a clear increasing trend in the first $\approx 10$ years (Figure 13e). Such a drift appears in the layer around 1-m deep and becomes stronger with depth (not shown). Because the same land model CLM4 is used in this study and in the CCSM4 historical simulation, this adjustment is likely a response to different land model resolutions and different atmospheric state. Similarly, the global-mean soil ice in the rcp85 experiment shows a steep decline in the first $\approx 10$ years (after the two-year spinup), followed by a still decreasing but weaker trend afterwards (Figure 13f). It is not clear that the weaker trend after 10 years represents the response to the future transient forcing or still converging to the model's own equilibrium state.

Spatially, most of the soil ice is stored over the northern high-latitudes and the Tibetan plateau, therefore the spin-up drift only exists in the limited regions that are in the coarse-resolution domain in our VR grids (Figure 14). Previous land modeling studies on permafrost regions suggest time scales of a hundred years for the water and energy cycles (Elshamy et al., 2020; Ji et al., 2022), especially with the extended bedrock layers down to $\approx 50$ m deep in CLM4 (Lawrence et al., 2008, 2012). We note that the global mean temperature of the bottom bedrock layer keep increasing throughout the rcp85 experiment with the overall increase of $\approx 3$ K over the 20 year period (not shown).

These results and previous findings suggest that high-latitude and high-altitude (e.g. Tibetan Plateau) soil hydrology and thermodynamics in the deep layers require decades to centuries of spinup. It is not clear whether the model adjustment in such deep, remote soil state can affect the simulated climate within the target refinement region. If such a remote effect exist, then it is necessary for global VR models to spin up the high-latitude/altitude soil state for a downscaling experiment, although it may be relevant to regional models in NA-CORDEX for the northern part of the domain. More detailed investigations are required on the coupling between the deep soil over the high-latitudes and the target downscaling region resulting from global teleconnection (e.g., by affecting meridional temperature gradient).



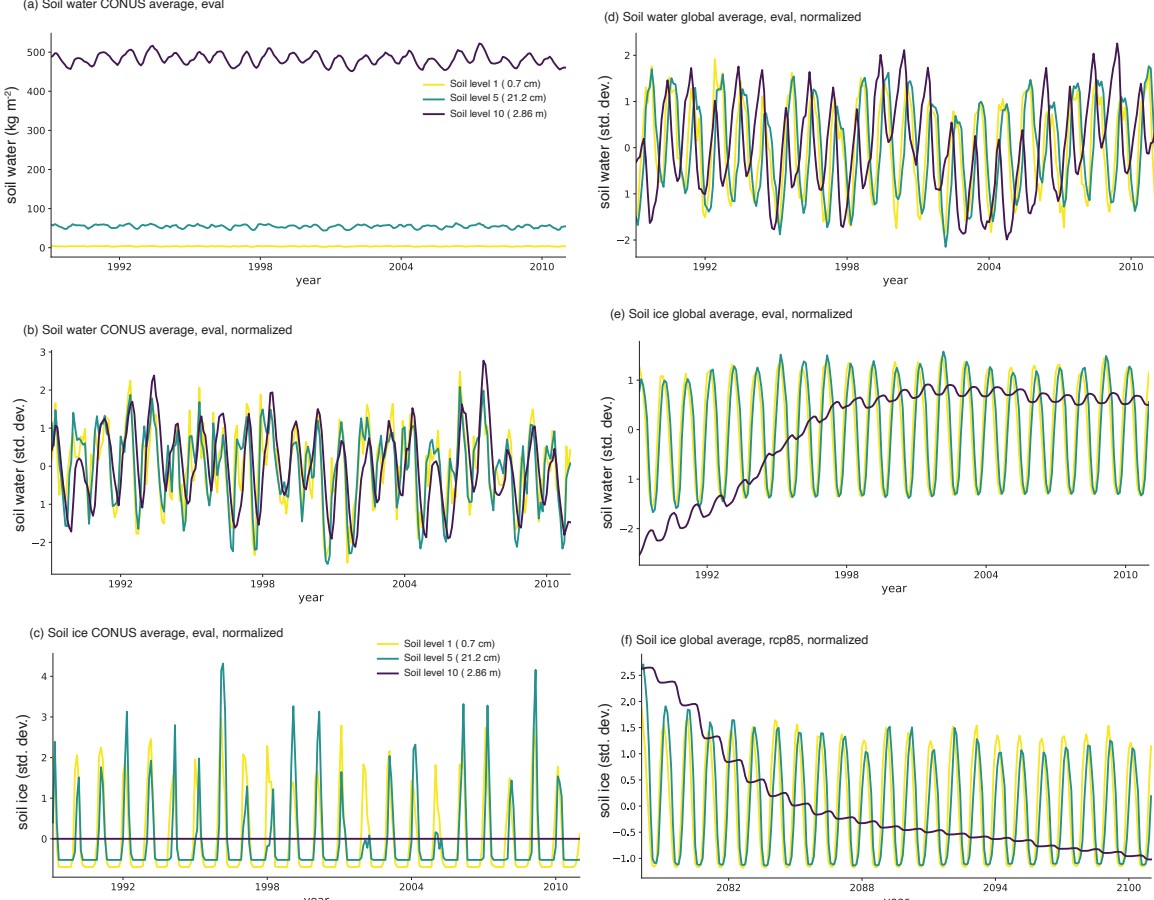

**Figure 13.** Time series of monthly mean (a) soil liquid water in the first, fifth, and tenth model layers averaged over the CONUS region in the VR50-200 eval experiment, (b) same as (a) but normalized by subtracting the temporal mean and dividing by the standard deviations, (c) normalized soil ice content averaged over CONUS, (d) globally averaged and normalized soil liquid content, (e) globally averaged and normalized soil ice content, and (f) same as (e) but from the rcp85 experiment.

# 6 Conclusions

The HyperFACETS project includes a large multi-institutional team and an important stakeholder engagement component to support climate adaptation efforts across a wide range of sectors. The engagement suggested that a timely and comprehensive

documentation of a climate model and the model output dataset is important to meet the growing demand of climate scientists, impact assessment researchers, stakeholders, and regional and national climate assessment activities for well documented and curated regional climate datasets. The aim of this work is to provide such a documentation for a relatively new global VR



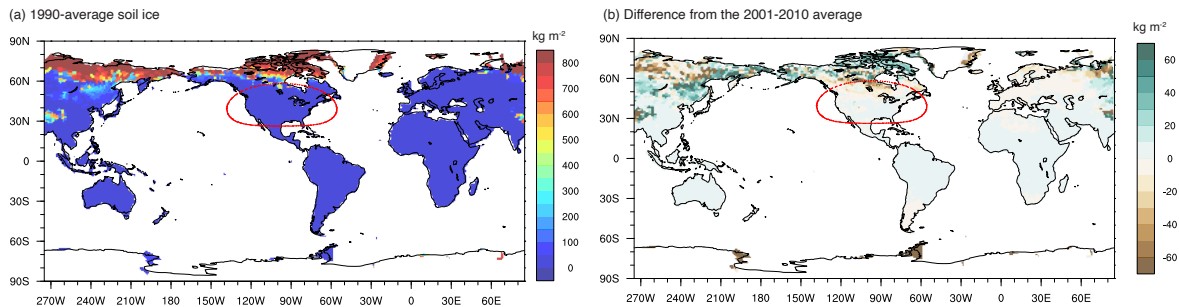

**Figure 14.** The soil ice content summed across soil layers in the VR50-200 eval simulation as (a) annual average over 1990 and (b) difference between the 1990 average (immediately after the spinup) and last-10 year average.

model framework, and to facilitate improvement in not only the model sciences and but also technical aspects of the climate model code and workflow from model configuration to post-processing under the changing HPC environment.

The CAM-MPAS simulations described in this paper are uniquely designed to facilitate the use and evaluation of the global VR model to complement the multi-model dynamical-downscaling products from the NA-CORDEX program and additional limited-area model simulations carried out under the HyperFACETS project. Details of the experimental CAM-MPAS model, downscaling simulations, output post-processing, data archive, and on-going improvement of the CAM-MPAS model are presented. A list of available variables and resources to analyze the raw model output on the unstructured grids are provided in

the appendices.

Model biases are described in the global scale (Sect. 5.2.1) and regional scale within the high-resolution domain of the VR simulations (Appendix F). It is noted that the biases are largely inherited from the CAM5.4 physics parameterizations, while some model sensitivities to resolutions and/or timestep lengths are different from those reported in previous studies using the CAM physics. The latter difference is mainly attributed to the resolution-dependent tuning of the convection timescale

parameter in the ZM deep convection scheme, highlighting a potential benefit of model tuning in VR downscaling. We expect that the model biases mentioned above will be reduced in the future CAM-MPAS (SIMA-MPAS) downscaling simulations coupled to the CAM6 physics parameterizations. A different deep convection parameterization, the Grell-Freitas scheme (Grell and Freitas, 2014), is being ported to SIMA-MPAS to alleviate several weaknesses in the CAM-MPAS VR configuration (Jang et al., 2022).

Looking ahead, an important next step would be to officially incorporate VR models into coordinated downscaling programs such as CORDEX (e.g., Prein et al., submitted to Climate Dynamics). Participation of VR models allows direct and more comprehensive intercomparison of limited-area and global VR models, but requires appropriate adaptations of the experimental protocol and analysis scope to address differences between the two modeling framework, such as the evaluation of soil state and large-scale circulations outside the refinement domain. Having both limited-area and VR models in a coordinated project



may also facilitate interactions between global and regional climate modeling communities, which could accelerate model development and workflow improvement to further reduce uncertainties in regional climate dataset.

*Author contributions.* KS, LRL, WJG, and LM designed the experiments and KS carried them out. SM guided post-processing of the MPAS output in accordance with the NA-CORDEX protocol. CMZ, WCS,and CZ contributed to porting the MPAS model into the CESM code. KS, CMZ, JJ, BEH, WCS, and AG regularly participated in monthly meetings to provide feedback on technical and scientific problems. CMZ,

JJ, WCS, AG, and CZ provided further help to solve technical problems. SL provided HPC support for the simulations and help obtaining usage data of the NERSC systems. KS prepared the manuscript with contributions from LRL, JJ, SM, BEH, WCS, WJG, and SL.

*Competing interests.* no competing interests are present

## 7  Code and data availability

Post-processed monthly and subset of daily variables ("essential" and "high priority" categories in the NA-CORDEX archive)

are available from the Pacific Northwest National Laboratory DataHub (Sakaguchi et al., 2022) (https://data.pnnl.gov/group/ nodes/dataset/13285). All the post-processed and raw model outputs are available from the NERSC High Performance Storage System (HPSS) through the NERSC Science Gateway Service (https://portal.nersc.gov/archive/home/k/ksa/www/FACETS/ CAM-MPAS).

The official version of the CEMS model is available as a public domain software from the project website (https://www.

cesm.ucar.edu/models/). The particular version of the experimental CAM-MPAS code used for this study is archived on Zenodo (Sakaguchi and Harrop, 2022) (https://doi.org/10.5281/zenodo.7262209). A set of input data files to reproduce the simulations reported here is available on Zenodo as well (Sakaguchi, 2022)(https://doi.org/10.5281/zenodo.7490129).

*Acknowledgements.* This work is funded by the U.S. Department of Energy, Office of Science, Office of Biological and Environmental Research program under Award DE- SC0016605 "A framework for improving analysis and modeling of Earth system and intersectoral

dynamics at regional scales" as part of the Regional and Global Model Analysis (RGMA) and MultiSector Dynamics (MSD) program areas. Some data analysis and collaborative work are also supported by the RGMA program area through the Water Cycle and Climate Extremes Modeling (WACCEM) scientific focus area. Chun Zhao was supported by the USTC Research Funds of the Double First-Class Initiative (YD2080002007) and the Strategic Priority Research Program of Chinese Academy of Sciences (XDB41000000). We acknowledge technical contributions from Michael Duda, Sang-Hun Park, and Peter Lauritzen to the CAM-MPAS code. This research used re-

sources of the National Energy Research Scientific Computing Center (NERSC), a U.S. Department of Energy Office of Science User Facility operated under Contract No. DE-AC02-05CH11231. Advice from the NERSC user support for resolving technical issues is greatly appreciated. The following software is used for processing model input and output data: netCDF Operators (NCO) version 4.7.0 (Zender





2017), NCAR Command Language (NCL) version 6.4 (UCAR/NCAR/CISL/TDD, 2017), GNU Parallel (Tange 2018),TaskFarmer (NERSC, https://docs.nersc.gov/jobs/workflow/taskfarmer/.



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



## Appendix A: Model input

The Figure A1 visualizes the input data flow for the CAM-MPAS model as explained in section 3.1. Most of the data preparation is to remap from the original input grids to the target MPAS grid, which is required for each different MPAS mesh.

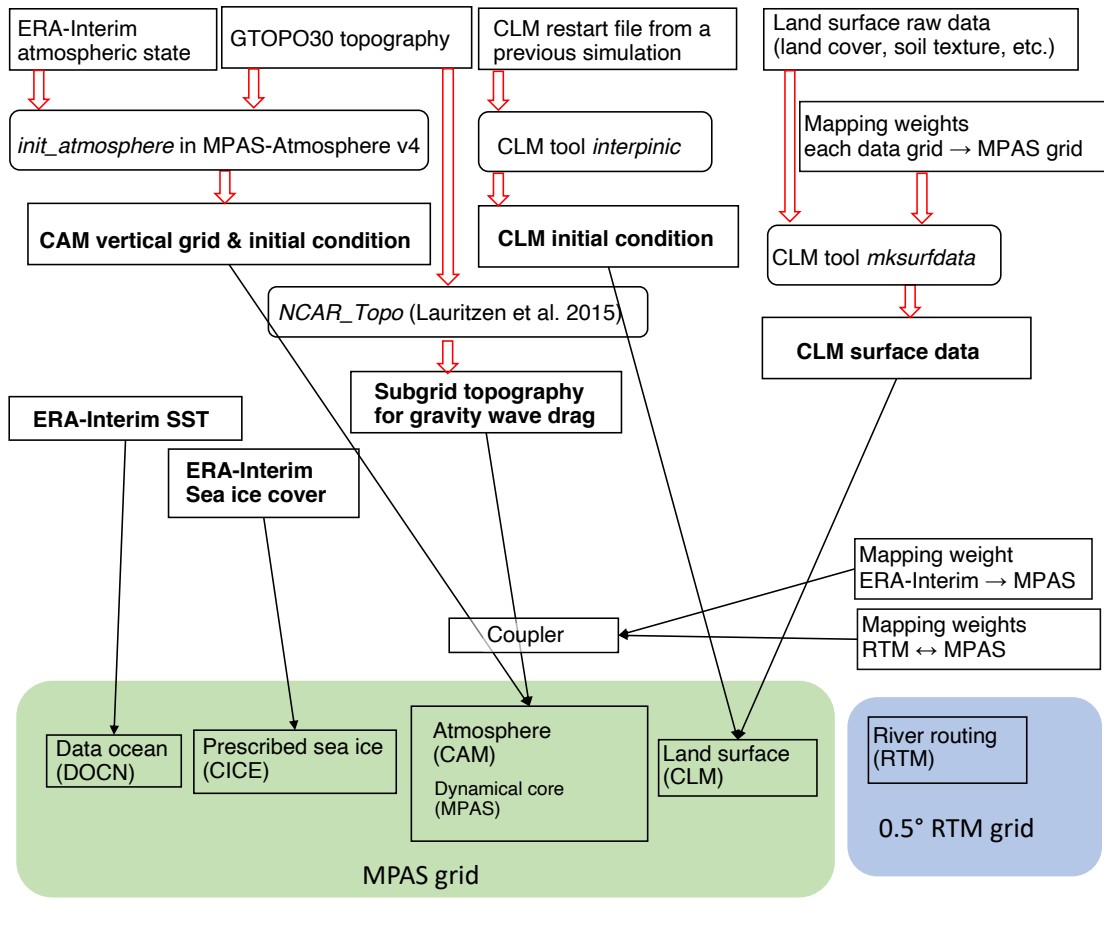

**Figure A1.** Input data flow described in 3. Note that the data flow in the current CESM2 model and future versions with the officially supported MPAS dynamical core are slightly different.

     Fortran namelist files that describe non-default model parameters, input data paths, output variables, and other model configurations, examples for VR25-100, are shared in a public space (https://portal.nersc.gov/cfs/m2645/pnnl/CAMMPAS/ namelists). A shell script (prod05_facets25-100_edison.sh) that executes a series of CESM scripts to set up CAM-MPAS 1080   VR25-100 is also available in the same directory.



## Appendix B: Raw model output

For users who prefer to analyze raw model outputs on the MPAS unstructured mesh, the raw outputs are available on the NERSC HPSS. Ancillary netcdf files for each MPAS grid are also available (Ancillary file link), including latitude/longitude coordinates of the grid cell centers, land/ocean masks, surface topography, and grid description in the SCRIP format (necessary to create remapping weights).

For users convenience, we provide a simple shell script to remap CAM-MPAS output to a regular latitude-longitude grid (regrid_CAM_MPAS_NCO.sh), and a jupyter notebook to calculate regional statistics on the raw MPAS grid data (RegionalAverage_mpasmesh.ipynb) in another directory (link). In the sub-directory "postprocess_6hr/", users can find more involved example with an ncl script to post-process six-hourly model outputs into the CORDEX format as well as shell scripts to run the ncl script in parallel on a NERSC KNL compute node using gnu parallel (Tange, 2018).

The MPAS mesh structure and other descriptions of the MPAS-Atmosphere model are provided in the MPAS user guide (Duda et al., 2019) and MPAS tutorial practice guide. A number of example python and NCL scripts to visualize data on the MPAS's unstructured grids are provided from the MPAS model website. To apply them to CAM-MPAS outputs, two adjustments are needed. First, variable names are different between the MPAS-Atmosphere and CAM-MPAS, the latter variable names can be found in the CAM documentation webpage (e.g., https://www.cesm.ucar.edu/models/cesm1.0/cam/docs/ug5_0/hist_flds_fv_cam5.html). Second, the dimension name "nCells" is used for variables defined at cell centers in MPAS-Atmosphere, while the dimension name is "ncol" in CAM-MPAS.

A raw CAM-MPAS output, or history file, contains multiple variables at one or more time records (up to 24), as opposed to a post-processed file that contains a single variable over a long period of time, from one year to the whole simulation period. All variables in the CAM-MPAS history files are either defined at or interpolated from cell edges to to cell centers. Readers are referred to the readme file https://portal.nersc.gov/cfs/m2645/pnnl/CAMMPAS/README_history.md) for details of the history file format, organization, variables, etc.





## Appendix C: Archived variables

List of variables available on the NA-CORDEX regional grid.

Table C1: Monthly variables.

| No. | Name | Long name | units |
|---|---|---|---|
| 1 | clh | High Level Cloud Fraction | fraction |
| 2 | cll | Low Level Cloud Fraction | fraction |
| 3 | clm | Mid Level Cloud Fraction | fraction |
| 4 | clt | Total Cloud Fraction | fraction |
| 5 | evspsbl | Evaporation | kg/m2/s |
| 6 | hfls | Surface Upward Latent Heat Flux | W/m2 |
| 7 | hfss | Surface Upward Sensible Heat Flux | W/m2 |
| 8 | hur500 | Relative Humidity at 500 mbar pressure surface | percent |
| 9 | hur700 | Relative Humidity at 700 mbar pressure surface | percent |
| 10 | hur850 | Relative Humidity at 850 mbar pressure surface | percent |
| 11 | hurs | Near-Surface Relative Humidity | percent |
| 12 | hus500 | Specific Humidity at 500 mbar pressure surface | fraction |
| 13 | hus700 | Specific Humidity at 700 mbar pressure surface | fraction |
| 14 | hus850 | Specific Humidity at 850 mbar pressure surface | fraction |
| 15 | huss | Near-Surface Specific Humidity | fraction |
| 16 | pr | Precipitation | m/s |
| 17 | prc | Convective Precipitation | m/s |
| 18 | prw | Water Vapor Path | kg/m2 |
| 19 | ps | Surface Air Pressure | Pa |
| 20 | psl | Sea Level Pressure | Pa |
| 21 | rlds | Surface Downwelling Longwave Radiation | W/m2 |
| 22 | rlus | surface upwelling longwave radiation | W/m2 |
| 23 | rlut | TOA Outgoing Longwave Radiation | W/m2 |
| 24 | rsds | Surface Downwelling Shortwave Radiation | W/m2 |
| 25 | rsdt | TOA Incident Shortwave Radiation | W/m2 |
| 26 | rsus | surface upwelling shortwave radiation | W/m2 |
| 27 | rsut | TOA outgoing shortwave radiation | W/m2 |
| 28 | sfcWind | Near-Surface Wind Speed | m/s |





| No. | Name | Long name | units |
|---|---|---|---|
| | | Table C1 – continued from previous page | |
| 29 | sic | Sea Ice Area Fraction | fraction |
| 30 | ta200 | Air Temperature at 200 mbar pressure surface | K |
| 31 | ta500 | Air Temperature at 500 mbar pressure surface | units |
| 32 | ta700 | Air Temperature at 700 mbar pressure surface | units |
| 33 | ta850 | Air Temperature at 850 mbar pressure surface | units |
| 34 | tas | Near-Surface Air Temperature | K |





Table C2: Daily variables. The variables in bold font are considered "essential" and "high priority" in NA-CORDEX and available from both PNNL Datahub and NERSC Science Gateway.

| No. | Name | Long name | units |
|---|---|---|---|
| 1 | **hurs** | Near-Surface Relative Humidity | percent |
| 2 | hus850 | Specific Humidity at 850 mbar pressure surface | fraction |
| 3 | **huss** | Near-Surface Specific Humidity | fraction |
| 4 | **pr** | Precipitation | m/s |
| 5 | prw | Water Vapor Path | kg/m2 |
| 6 | **ps** | Surface Air Pressure | Pa |
| 7 | psl | Sea Level Pressure | Pa |
| 8 | **sfcWind** | Near-Surface Wind Speed | m/s |
| 9 | ta200 | Air Temperature at 200 mbar pressure surface | K |
| 10 | ta500 | Air Temperature at 500 mbar pressure surface | K |
| 11 | ta850 | Air Temperature at 850 mbar pressure surface | K |
| 12 | **tas** | Near-Surface Air Temperature | K |
| 13 | **tasmax** | Daily Maximum Near-Surface Air Temperature | K |
| 14 | **tasmin** | Daily Minimum Near-Surface Air Temperature | K |
| 15 | ua200 | Eastward Wind at 200 mbar pressure surface | m/s |
| 16 | ua850 | Eastward Wind at 500 mbar pressure surface | m/s |
| 17 | **uas** | Eastward Near-Surface Wind (lowest model level) | m/s |
| 18 | utmq | Vertically Integrated Eastward Water Vapor Flux | kg/m/s |
| 19 | va200 | Northward Wind at 200 mbar pressure surface | Pa |
| 20 | va850 | Northward Wind at 850 mbar pressure surface | Pa |
| 21 | **vas** | Northward Near-Surface Wind (lowest model level) | m/s |
| 22 | vtmq | Vertically Integrated Northward Water Vapor Flux | kg/m/s |
| 23 | wap500 | Omega (=dp/dt) at 500 mbar pressure surface | Pa/s |
| 24 | zg200 | Geopotential Height at 200 mbar pressure surface | m |
| 25 | zg500 | Geopotential Height at 500 mbar pressure surface | m |




Table C3: 6-hourly variables.

| No. | Name | Long name | units |
|-----|------|-----------|-------|
| 1 | hurs | Near-Surface Relative Humidity | percent |
| 2 | hus850 | Specific Humidity at 850 mbar pressure surface | fraction |
| 3 | huss | Near-Surface Specific Humidity | fraction |
| 4 | pr | Precipitation | m/s |
| 5 | prw | Water Vapor Path | kg/m2 |
| 6 | ps | Surface Air Pressure | Pa |
| 7 | psl | Sea Level Pressure | Pa |
| 8 | sfcWind | Near-Surface Wind Speed | m/s |
| 9 | ta200 | Air Temperature at 200 mbar pressure surface | K |
| 10 | ta500 | Air Temperature at 500 mbar pressure surface | K |
| 11 | ta850 | Air Temperature at 850 mbar pressure surface | K |
| 12 | tas | Near-Surface Air Temperature | K |
| 13 | tasmax | Daily Maximum Near-Surface Air Temperature | K |
| 14 | tasmin | Daily Minimum Near-Surface Air Temperature | K |
| 15 | ua200 | Eastward Wind at 200 mbar pressure surface | m/s |
| 16 | ua850 | Eastward Wind at 500 mbar pressure surface | m/s |
| 17 | uas | Eastward Near-Surface Wind (lowest model level) | m/s |
| 18 | utmq | Vertically Integrated Eastward Water Vapor Flux | kg/m/s |
| 19 | va200 | Northward Wind at 200 mbar pressure surface | Pa |
| 20 | va850 | Northward Wind at 850 mbar pressure surface | Pa |
| 21 | vas | Northward Near-Surface Wind (lowest model level) | m/s |
| 22 | vtmq | Vertically Integrated Northward Water Vapor Flux | kg/m/s |
| 23 | wap500 | Omega (=dp/dt) at 500 mbar pressure surface | Pa/s |
| 24 | zg200 | Geopotential Height at 200 mbar pressure surface | m |
| 25 | zg500 | Geopotential Height at 500 mbar pressure surface | m |



**Table C4.** 3-hourly variables

| No. | Name | Long name | units |
|-----|--------|--------------------------------------------|------------|
| 1 | tas | Near-Surface Air Temperature | K |
| 2 | tasmax | Daily Maximum Near-Surface Air Temperature | K |
| 3 | tasmin | Daily Minimum Near-Surface Air Temperature | K |
| 4 | pr | Precipitation | m s$^{-1}$ |
| 5 | ps | Surface Pressure | Pa |





## Appendix D: Computation at NERSC

The Cori Haswell (HW) and Edison systems at NERSC feature the same processor family (Intel Xeon processor) on more traditional, massively parallel distributed memory architectures with fewer cores of higher CPU frequencies and larger memory per node. In contrast, Cori Knights-Landing (KNL) employs an architecture with a different parallelism philosophy of many-cores, wider-vector units, and non-uniform and high bandwidth memory access with the Intel Xeon Phi processors (He et al., 2018). The transition to many-core architecture has occurred in multiple HPC facilities, motivated by better energy efficiency (Allen et al., 2018; Loft, 2020) and preparation of user applications for more extreme many-core architecture with GPU systems (NERSC, 2014).

To compile the CAM-MPAS code on all of the NERSC systems mentioned above, we use the Intel compiler wrapper provided by the system vendor Hewlett Packard Enterprise Cray (NERSC, 2018). The libraries and compilers we used can be seen in the file

[top directory]/cime/cime_config/cesm/machines/config_machines.xml

within the source code directory tree. All simulations except for "UR120-new" use the same model code and same compiler options. The compiler optimization level (O1) is lower than the default for CESM (O2) for the CAM-MPAS code being experimental. The UR120-new is run with the beta-version of the MPASv6-CESM2 coupled code with the O2 optimization, which can improve simulation throughput by up to $\approx 10$ % compared to O1 based on our benchmark simulations on KNL. Shared-memory parallelism is not used because the MPAS-Atmosphere version 4 does not support OpenMP, and the CESM code does not necessarily show better performance with the hybrid OpenMP + MPI compared with MPI-only configurations (Helen He, personal communication). For MPI-only jobs, Heinzeller et al. (2016) recommended 100–150 grid columns per MPI task to achieve good throughput for the stand-alone MPAS-Atmosphere model. Not all of our node configurations follow their recommendation because of the reasons mentioned below.

At NERSC, queue wait time depends on requested wall-clock hours and number of nodes, but the former tends to be more important than the latter (Figure D1). Therefore, we aimed for wall-clock time of 5-6 hours or less to integrate one- to six-months in a single job to avoid a long queue wait time. Then we looked for sufficient numbers of MPI tasks to achieve this goal to finally determine the number of nodes to request for production simulations. We were also interested in comparing different systems during the transition period from Edison to Cori, so some simulations used similar numbers of nodes or MPI tasks on different systems.

We explored several reasons for the lower throughput of our CAM-MPAS code on Cori-KNL than on Cori-HW and especially the older system Edison. Primary reasons seem to be inefficient memory usage, under-usage of shared memory parallelism, and source code style that is not easily vectorized by compilers (in addition to the lower level of compiler optimization we chosen as mentioned above). As summarized by He et al. (2018) and Barnes et al. (2017), the previous system Edison has two Intel Ivy Bridge 2.6 GHz 12-core CPUs (24 cores per node) and 64 GB memory with ∼100 GB/s bandwidth on each node. Cori-KNL, on the other hand, has one Xeon-Phi 7250 1.4 GHz processor that has 68 physical cores, each of which can be used with 4 hardware threads. A KNL node has a larger 96 GB memory with slower 85 GB/s bandwidth than Edison, but also





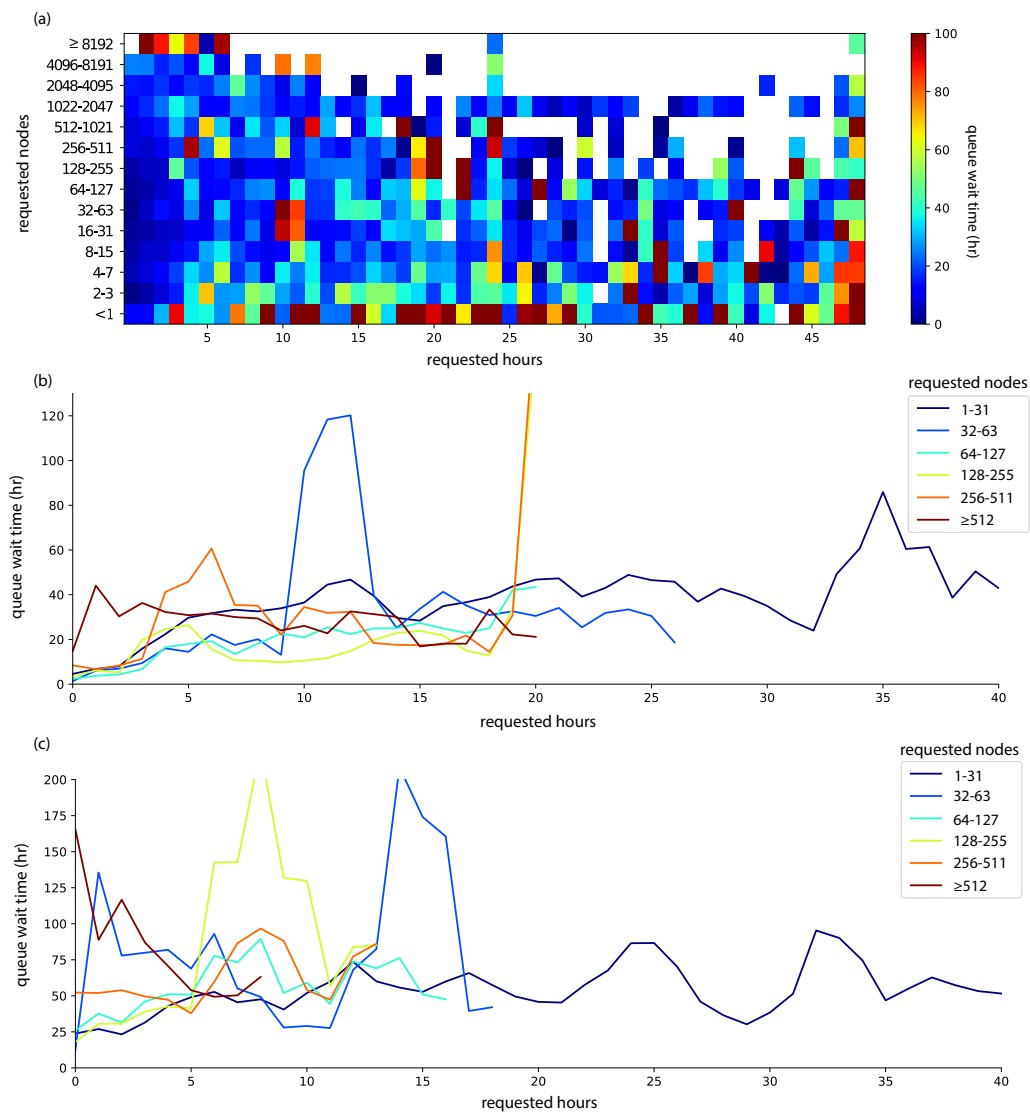

**Figure D1.** The year 2020 annual average queue wait time for (a) Cori KNL as a function of requested wall-clock hours (x-axis) and requested number of nodes (y-axis), (b) same as (a) but as line plots by further averaging the bins of requested hours to six groups as shown in the legend, (c) same as (b) but for Cori HW system. Note different ranges of the y-axes in (b) and (c). The data was obtained from the "MyNERSC" website (accessible only by NERSC users) with the help from the NERSC user support.

provides additional 16 GB high-bandwidth (450 GB/s) memory. Despite the lower clock frequency, KNL's Xeon-Phi processor
performs 32 double precision floating point operations per second (FLOPS) per cycle compared to 8 FLOPS per cycle by Edison's Ivy Bridge processor.



The overall performance of climate model code is typically limited by memory latency and bandwidth rather than arithmetic speed (e.g., Fuhrer et al., 2018; Dennis et al., 2019), except for some components such as the MG2 microphysics (Barnes et al., 2017). A naive use of all the 68 cores on KNL nodes as MPI ranks lead to 0.5 GB memory per rank (using KNL's two different memory units as a single entity), about one-fifth of 2.7 GB per rank when using 24 MPI ranks per node on Edison. In addition to this memory-per-rank difference, we found inefficient memory use by the CESM1.5 code, which became clear with very high resolution (more than 1 million columns) but already impacted resolutions with $\approx 0.5$ million grid columns, including the VR12-46 and UR30 grids. An example of the inefficient memory use is to store unnecessarily long arrays on the memory of each node (e.g., those cover the whole global domain instead of the sub-domain assigned to the MPI rank), which exacerbates smaller memory per MPI rank on the NKL node.

Recommended programming models for Cori KNL are vectorization, shared-memory parallelism, and control of data block size within the 16 GB high-speed memory. It was found that such programming design is not very common within the CESM code during the NERSC Exascale Science Applications Program (NESAP), which was established to help NERSC users to optimize their applications for KNL (He et al., 2018). As part of the NESAP, two sub-components of the CESM model were optimized by the code developers and NERSC support staff. The MG2 microphysics code was found to be bounded by computation with poor vectorization, and improved code structure for easier vectorization enhanced its speed by about 75% (He, 2016; Barnes et al., 2017). Optimizations of the High Order Methods Modeling Environment (HOMME) dynamical core involved both better vectorization and rewriting the OpenMP loops, which together achieved twice faster performance on KNL (Barnes et al., 2017; Dennis et al., 2019). While some optimizations are more specific to KNL, many of the code changes improve performance on other systems such as the Cheyenne system in the National Center for Atmospheric Research - Wyoming Supercomputing Center (Dennis et al., 2019).

It is generally difficult for these specific optimizations to be incorporated into the official release of the CESM code (let alone off-branched experimental versions) within the life time of a typical HPC system of 4–5 years. This can be a serious and common challenge for climate modeling research groups, whose numerical experiments require long simulation time. Fortunately, the MPAS-Atmosphere code went through several optimizations in version 5, including changes similar to those reported in the above studies. In addition, the memory-scaling issues in the CAM code have been addressed in the current version of CESM2. Along with other numerous changes from CESM1.5 to CESM2.1, the latest version of CAM-MPAS achieves substantially better performance on KNL (Table. 5).





## Appendix E: Global climate

This appendix provides additional information for the global climate and its resolution sensitivity in the CAM-MPAS simulations.

### E1  Present-day climate biases and resolution sensitivities

As mentioned in the main text (section 5.2.1), the present-day climatology of near-surface air temperature (TAS) is similar across the resolutions (Figure E1). All the CAM-MPAS eval simulations share regional biases, most notably the warm bias in

the mid-latitude continents and in the Southern Hemisphere storm tracks. On the other hand, TAS over complex terrains such as Tibetan Plateaus and western Americas show visible difference across resolution.

Previous studies of the CAM model suggest that the too warm TAS in the Southern Hemisphere storm track is related to the underestimated low-level liquid clouds (Bogenschutz et al., 2018) and overestimated wind speed (and associated vertical mixing) in the lower atmosphere. For the Arctic region, the CAM5 physics was shown to underestimate Arctic clouds, leading

to less downward longwave radiation, smaller surface net energy, and colder surface temperature (English et al., 2014; McIlhattan et al., 2017). Note that the sea-ice model in the CESM AMIP configuration interactively calculates the surface energy balance and temperature given the prescribed ice coverage, unlike the open ocean surface where the surface skin temperature is prescribed (section 3).

The contour plots of precipitation biases against GPCP show greater variations among simulations than TAS (Figure E2).

UR120 shows the smallest regional bias across the globe, presumably because its grid resolution and timestep are close to those of FV1°, to which we tuned the CAM5.4 physics with the prescribed aerosol scheme (section 2.2).

### E2  Future climate changes and resolution sensitivities

Figure E4 compares mean precipitation changes from the historical to RCP8.5 periods in the five CAM-MPAS simulations and MPI-ESM-LR simulation. As mentioned in the main text, the overall spatial patterns are similar across the simulations.

Because the contour color range is set for the larger changes over the ocean, Figure E5 masks the ocean grid points and focuses on land.



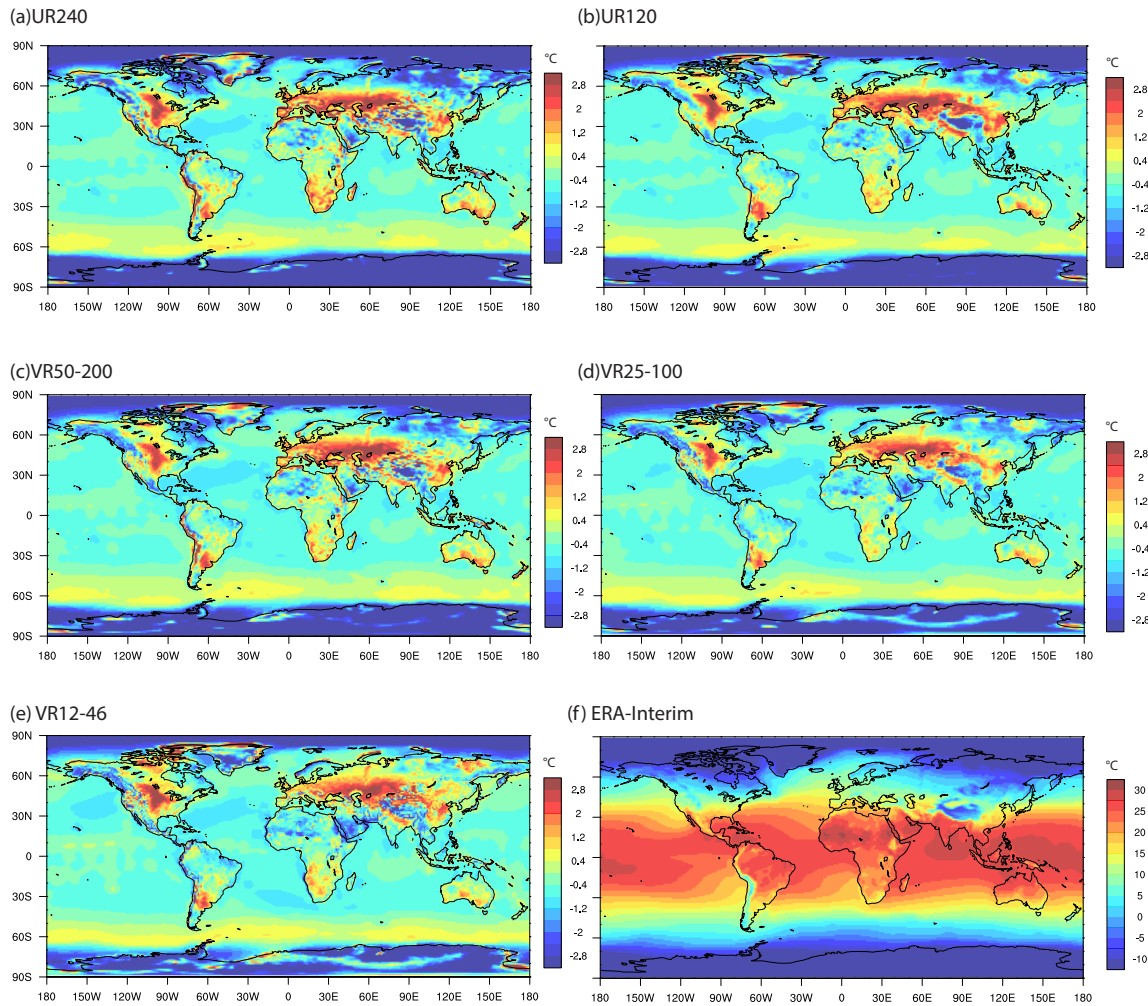

**Figure E1.** Difference of climatological 2-m air temperature over the 1990-2010 (2001-2010 for VR12-46) period between CAM-MPAS simulations and ERA-Interim (a)-(e), and the annual mean temperature in ERA-Interim (f). The ERA-Interim sea surface temperature and sea ice cover are used as input for the CAM-MPAS AMIP simulations.





**Figure E2.** Difference of climatological surface precipitation over the 1997-2010 (2001-2010 for VR12-46) period between CAM-MPAS simulations and GPCP (a)-(e) and the annual mean precipitation in GPCP (f). Grid imprinting in the contour plots (a) and (c) over some regions (e.g., the Indian Ocean west of Africa) is a result of conservatively remapping coarser MPAS grids to the finer ≈ 0.7° latitude-longitude grid.





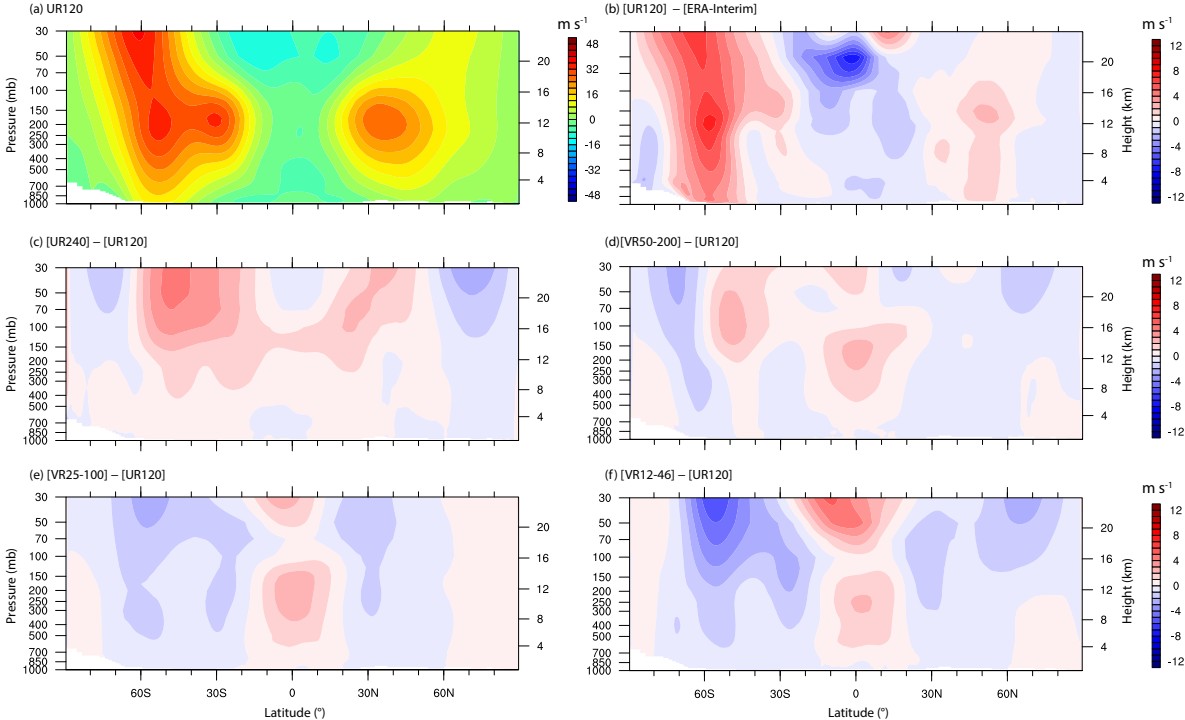

**Figure E3.** Annual climatology of zonal mean zonal wind in (a) UR120, (b) UR120 bias compared to ERA-Interim, and differences between UR120 and other CAM-MPAS resolutions (c) UR240, (d) VR50-200, (e) VR25-100, and (f) VR12-46.

**Table E1.** Observational dataset used in Table 6 and their reference, obtained through Atmospheric Model Working Group (2014).

| Name | Variables | Period | Reference |
|---|---|---|---|
| ISCCP | cloud fraction | 1983-2001 | Rossow and Schiffer (1999) |
| CLOUDSAT | cloud fraction | 1983-2001 | Marchand et al. (2008) |
| ERBE | energy flux and cloud radiative forcing | 1985-1989 | Smith et al. (1987) |
| CERES-EBAF | energy flux and cloud radiative forcing | 2000-2010 | Loeb et al. (2009) |
| GPCP | precipitation rate | 1979-2009 | Adler et al. (2003) |
| AIRS | precipitable water | 1988-1999 | Susskind et al. (2003) |
| NVAP | precipitable water | 1988-1999 | Randel et al. (1996) |
| MODIS | precipitable water | 2000-2004 | King et al. (2003) |
| ERA40 reanalysis | precipitable water | 1980-2001 | Uppala et al. (2005) |
| JRA25 reanalysis | precipitable water | 1979-2004 | Onogi et al. (2007) |
| ERA-Interim reanalysis | UA200, precipitable water | 1989-2005 | Dee et al. (2011) |
| HadCRUT3 | surface air temperature | 1961-1990 | Brohan et al. (2006) |



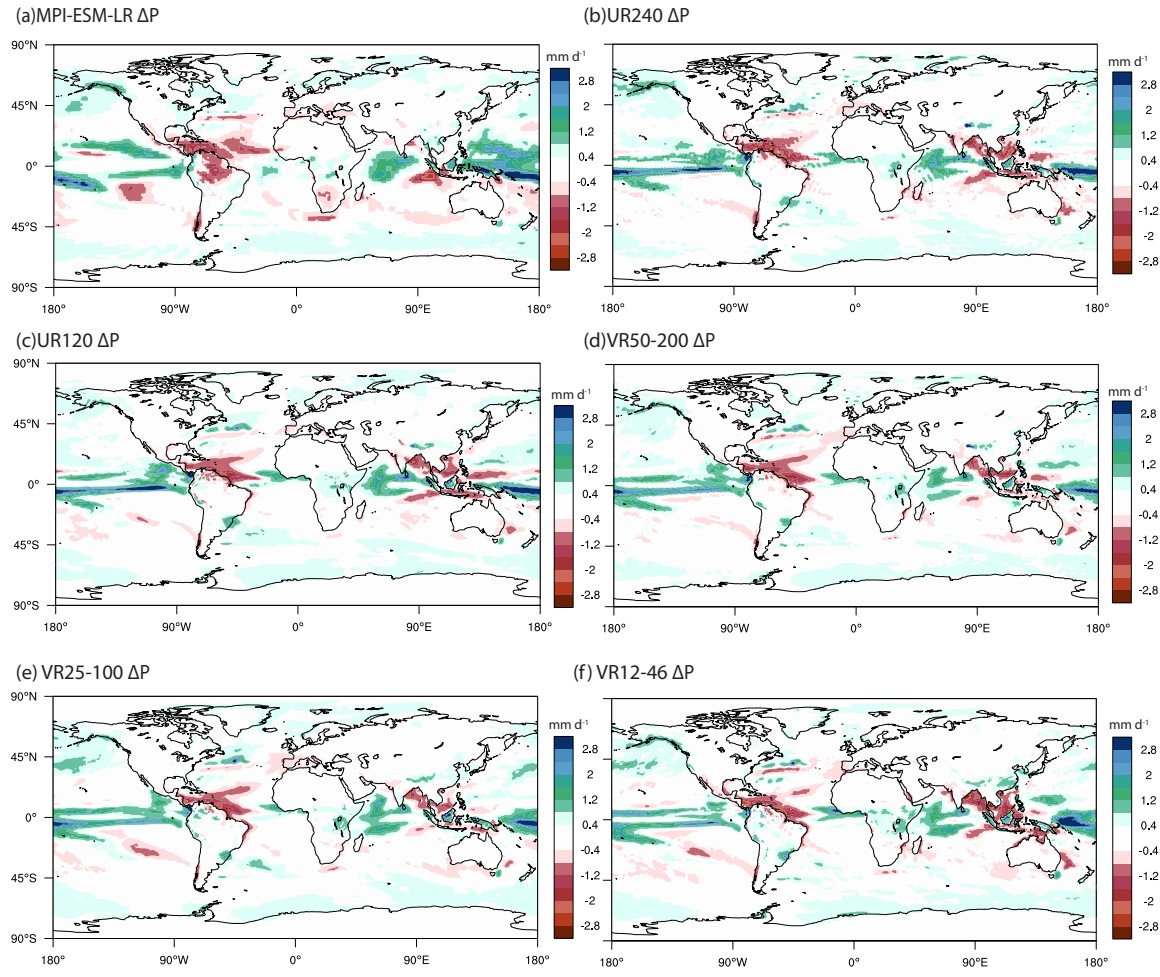

**Figure E4.** Projected precipitation change from the historical to RCP8.5 periods in (a) MPI-ESM-LR, (b) UR240, (c) UR120, (d) VR50-200, (e) VR25-100, and (f) VR12-46.



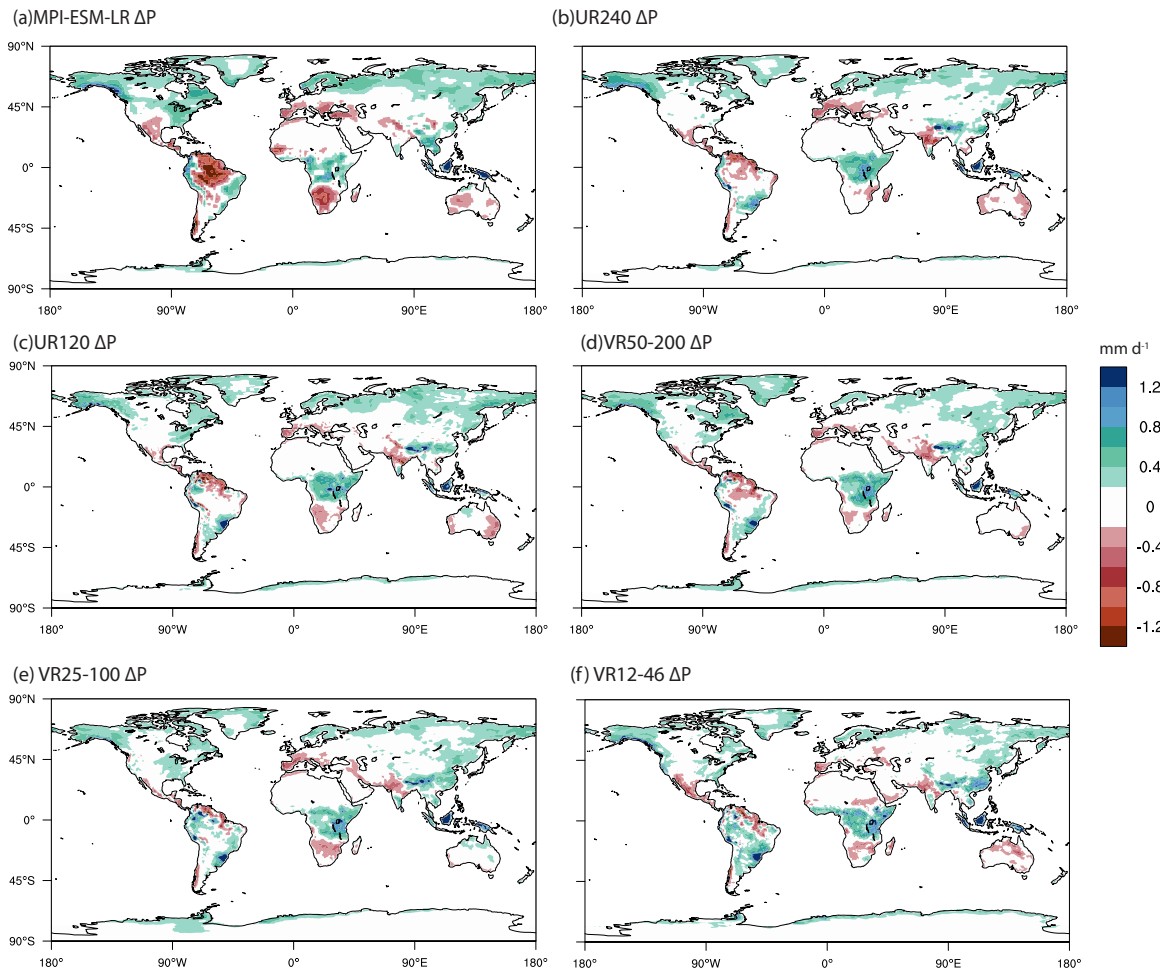

**Figure E5.** Same as Figure E4 but the ocean grid points are masked and a narrower color range is used to focus on precipitation change over land.



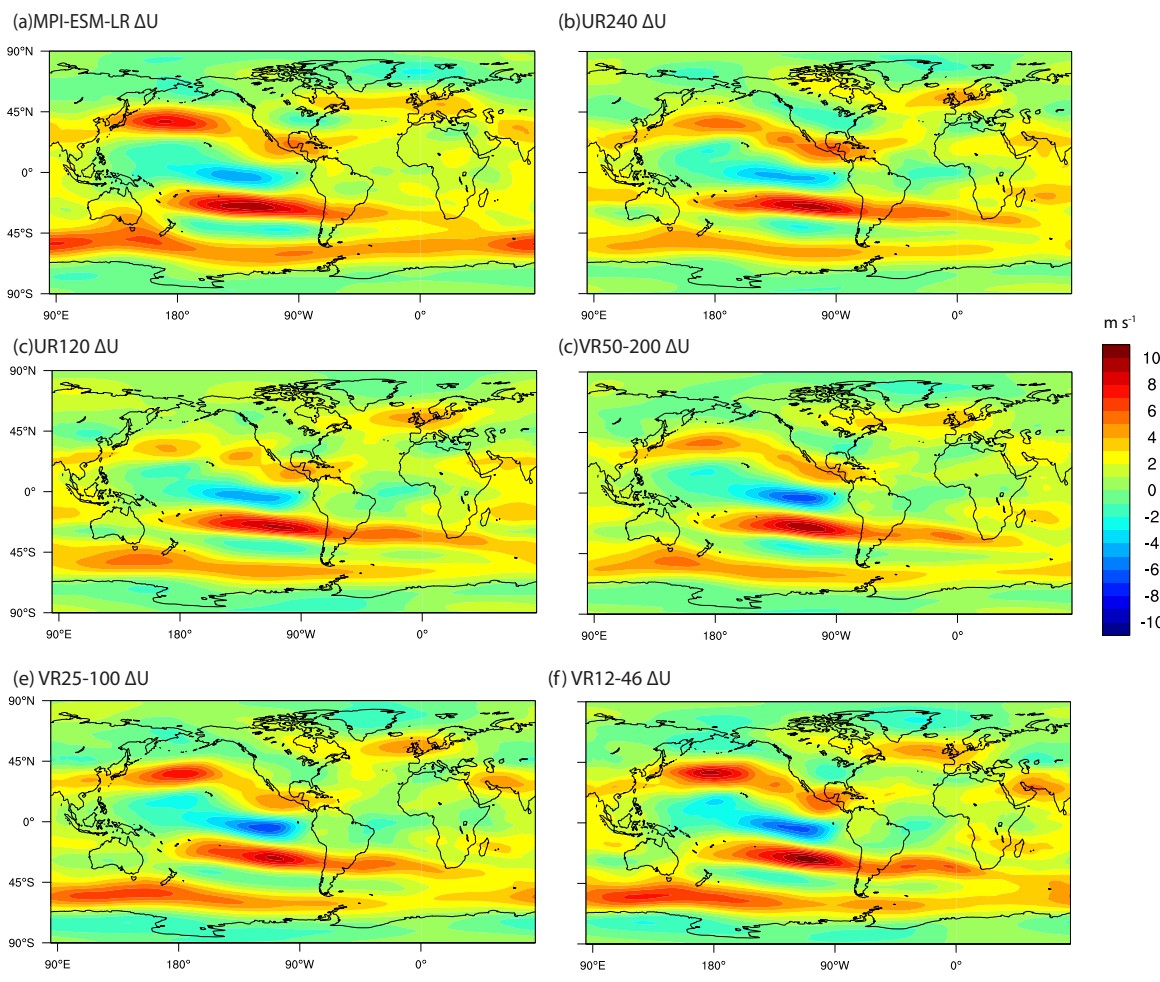

**Figure E6.** Same as Figure E4 but for the projected change of zonal wind at the 200 hPa level (△UA200).



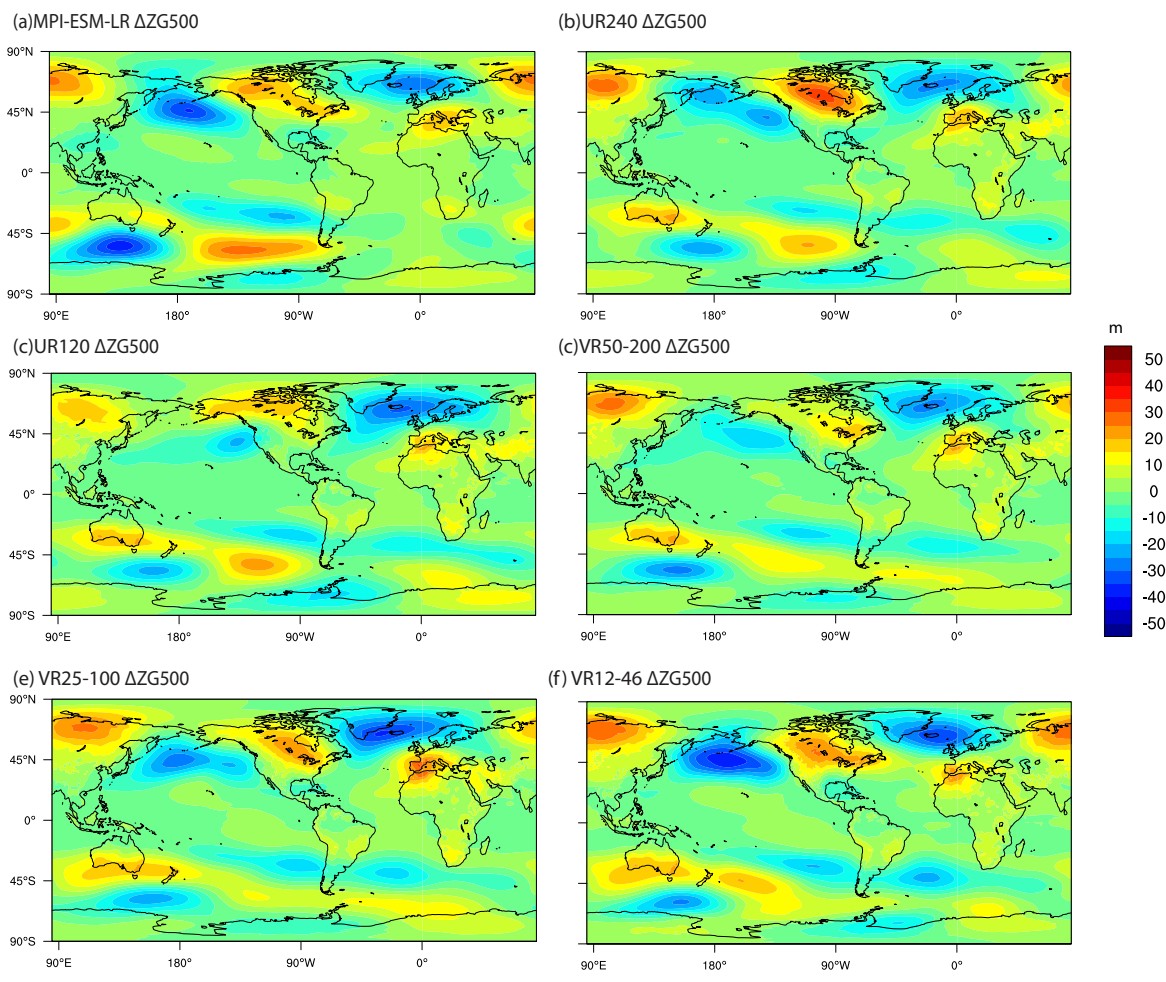

**Figure E7.** Same as Figure E4 but for the the projected change of zonal anomaly of geopotential height at the 500 hPa level (△ZG500).



## Appendix F: Regional climate

This appendix provides an overview of the regional climate evaluated over CONUS (defined as 105°–85°W, 30°–47°N) and its resolution sensitivity in the CAM-MPAS simulations.

The time series of annual- and regional-mean near-surface air temperature (TAS) over the CONUS region is nearly identical among the different resolutions (Figure F1a, b), except for some seasonal maxima and minima where significant differences can arise in some years. The TAS spatial patterns, shown as differences from the ERA-Interim temperature in Figure F2, illustrate that the spatial patterns of the biases (and TAS itself) are also similar across simulations over the central and eastern U.S. Over the western U.S., with complex surface topography, greater spatial variability is simulated by finer spatial resolutions. The topography-related spatial variability seems to be filtered out and does not affect the regional average time series.

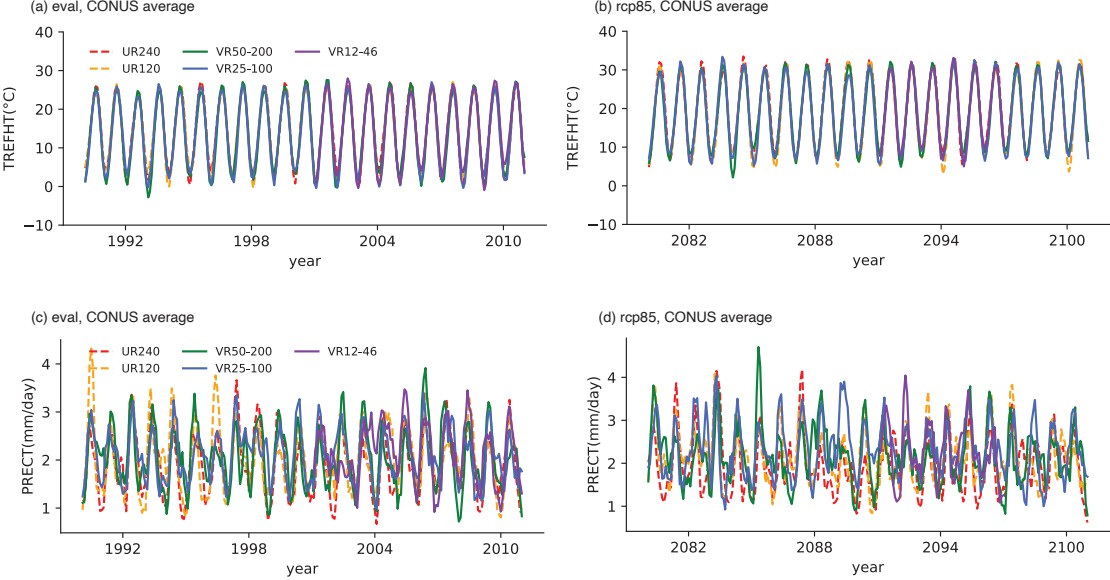

**Figure F1.** Time series of monthly average (a) near-surface air temperature in the present-day (eval) simulations, (b) near-surface air temperature in the future (rcp85) simulations, (c) precipitation in the present-day (eval) simulations, and (d) precipitation in the future (rcp85) simulations. All variables are averaged over the CONUS domain defined as 85–105°W, 30–47°N.

The CONUS-average precipitation, on the other hand, varies significantly across years and among the resolutions (Figure F1c, d). The resolution-sensitivity of the CONUS-average precipitation is not as simple or systematic as the global mean precipitation. A subtle but consistent increase with resolution appears in the total (convective plus large-scale) precipitation after further averaging over time, but not in individual convective and large-scale components (Table F1). As in TAS, the spatial patterns of precipitation bias are similar across simulations over the central and eastern U.S., but greater variability appears with higher resolution over the western U.S. (Figure F3).



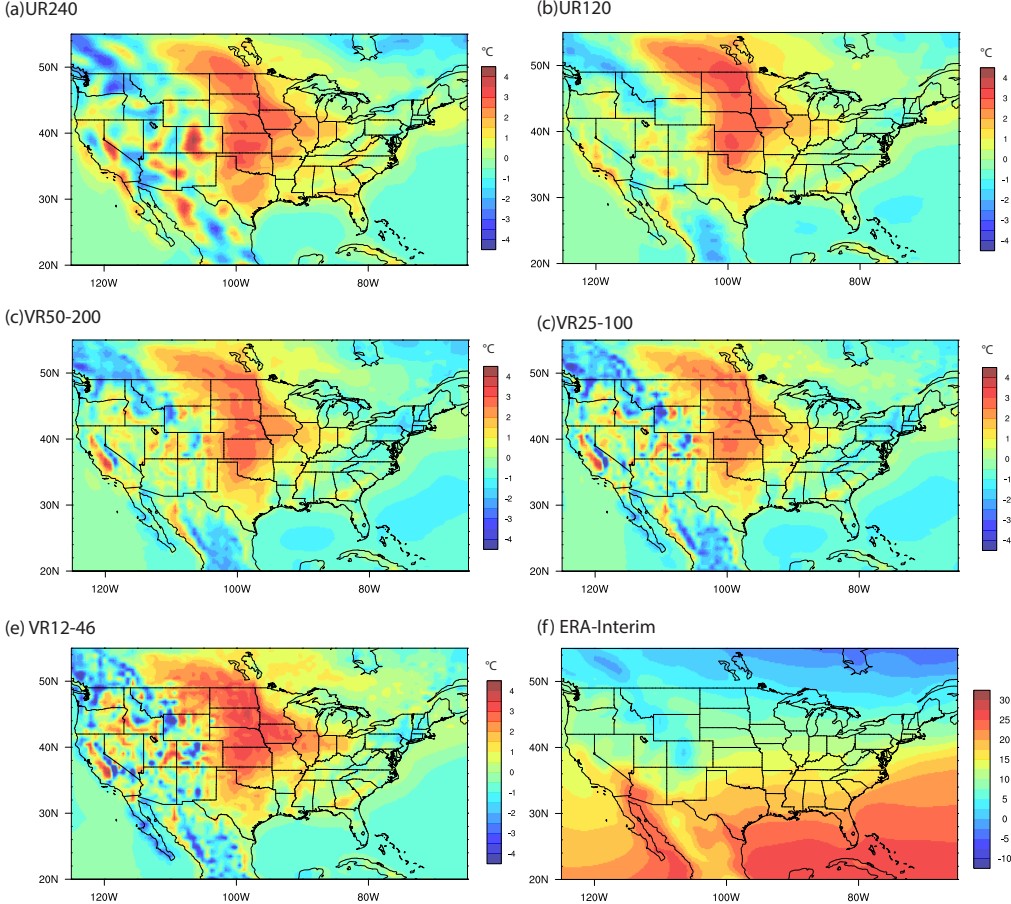

**Figure F2.** Annual mean 2m air temperature bias as in Figure E1 but showing only the CONUS region. The last panel (f) is not the bias but the mean surface temperature from ERA-Interim.

An interesting difference between the TAS and precipitation biases is that the TAS warm bias is maximized over the northern central US (35–55°N) and shows little sensitivity to resolution, while the precipitation dry bias is greatest in the southern central US (30–35°N) and does show sensitivity to resolution. Multiple performance metrics — the ratio of spatial variance, mean bias, and centered (i.e., mean bias already removed) root mean square error (CRMSE) — calculated over the CONUS region suggest that surface precipitation is best simulated by VR12-46 (Figure F4b). Comparing all resolutions, both the spatial variability (variance ratio and centered RMSE) and the mean (normalized bias) of precipitation are better simulated by finer resolution. On the other hand, the correlation and variance ratio for TAS depend more weakly on resolution (Figure F4a).

Other hydrological components show more consistent resolution-sensitivities than the surface precipitation. For example, the regional average cloud cover and low-level humidity become progressively smaller with higher resolution (Table F1). The



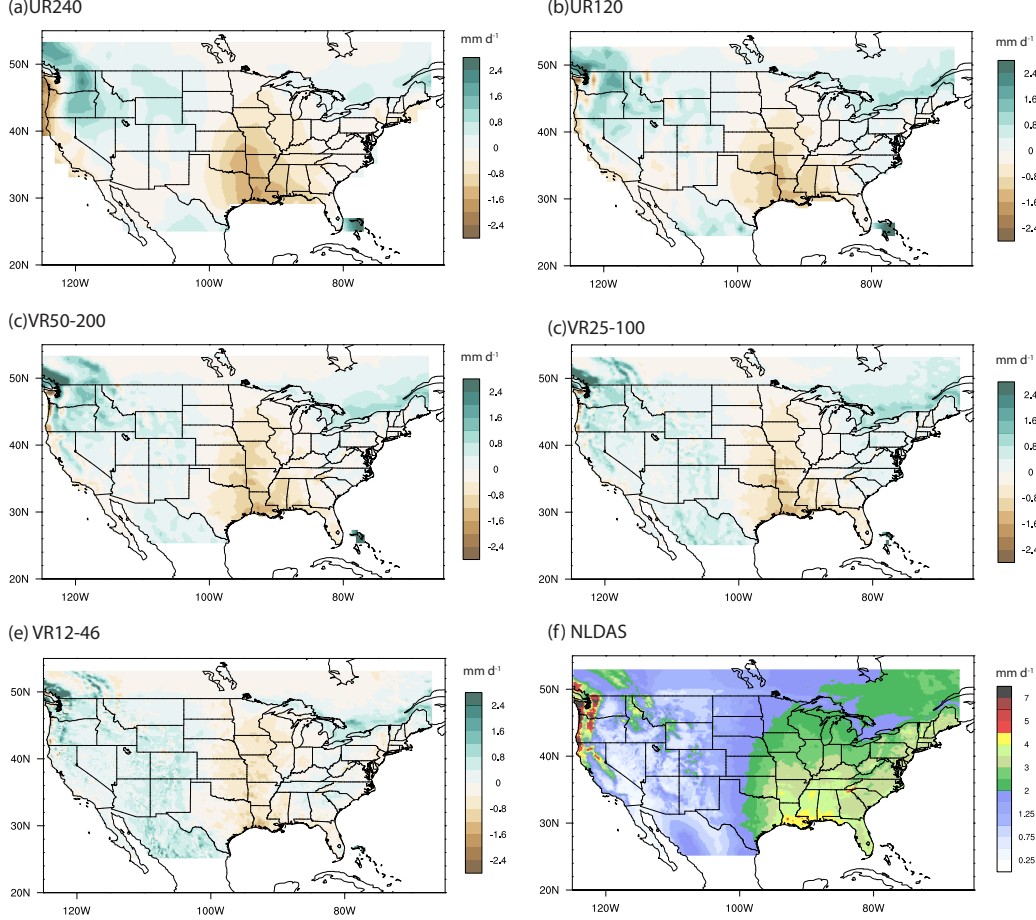

**Figure F3.** Annual mean precipitation bias as in Figure E1 but showing only the CONUS region and using NLDAS data instead of GPCP as reference (f).

resolution-sensitivity is more subtle for large-scale forcing terms such as relative humidity and meridional winds at the 850 hPa level (denoted as RH850 and VA850, respectively) and zonal wind at the 200 hPa level (UA200), which have been suggested to be important for the regional hydrological cycle over CONUS (e.g. Bukovsky et al., 2017; Song et al., 2019). Compared to ERA-Interim, VA850 metrics improve with finer resolution (Figure F4c). In contrast, UA200 and RH850 do not show similar

improvement with increasing spatial resolution. A lack of coherent resolution sensitivity of UA200 is consistent with Figure E3 where little difference is seen between the simulations in the Northern Hemisphere mid-latitudes.

As we prepare a more detailed documentation of the regional climate simulations, we refer potential data users to the following studies evaluating the aspects of the CAM-MPAS simulations not documented here. Feng et al. (2021) performed in-depth analysis of the simulated precipitation over CONUS, focusing on the mesoscale convective systems (MCSs) and

associated large-scale environment. They found that the model is capable of simulating the large-scale meteorological patterns



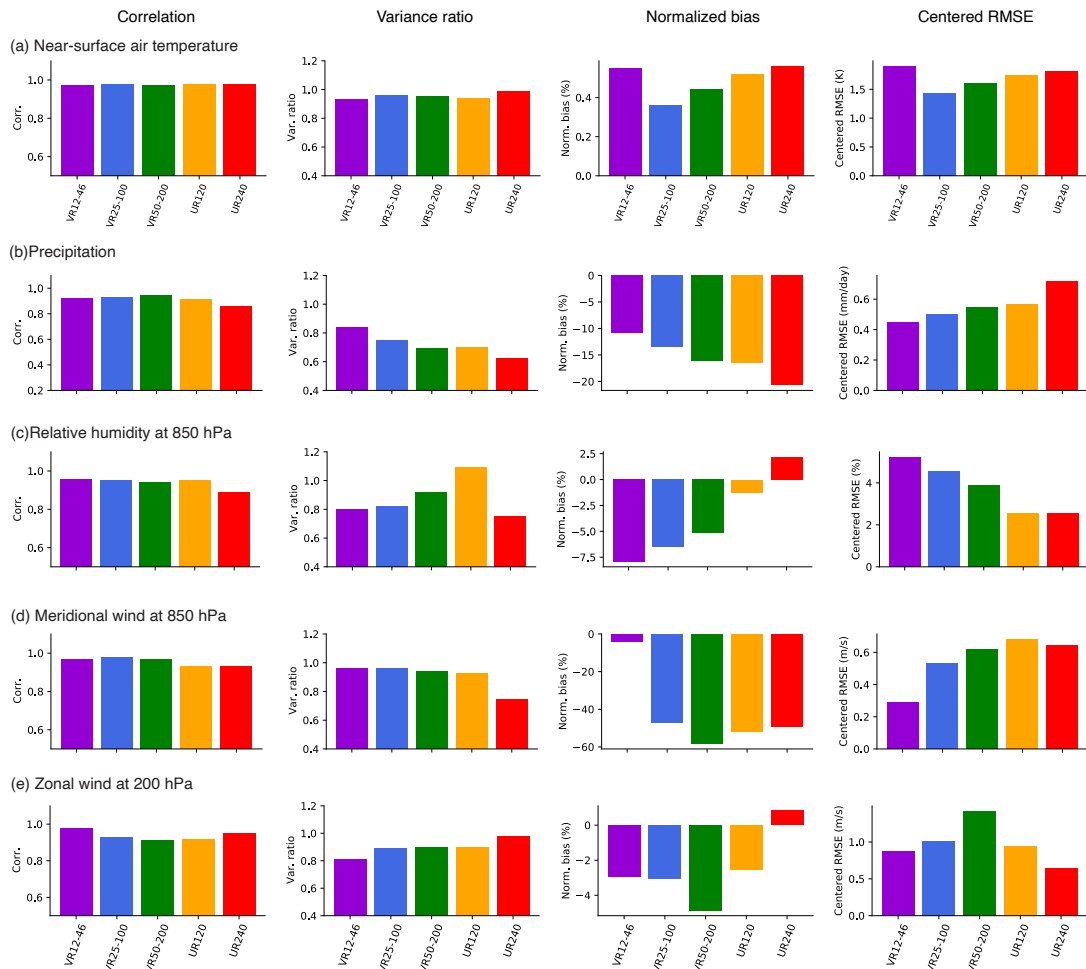

**Figure F4.** Model error metrics calculated over the CONUS region for (a) near-surface temperature, (b) precipitation, (c) relative humidity at 850 hPa, (d) meridional wind at 850 hPa, and (e) zonal wind at 200 hPa. Annual averages for the years 1990–2010 (2000–2010 for VR12-46)) from NLDAS is used as a reference for precipitation, and those from ERA-Interim are used for other variables. Four error metrics (as a column) are presented: 1) linear correlation of spatial patterns (Corr.), 2) ratio of the spatial variance ($\sigma^2_{ref}/\sigma^2_{mod}$, subscripts ref and mod refer to the reference data and model, respectively), 3) normalized bias (%) (($\bar{X}_{mod} - \bar{X}_{ref})/\bar{X}_{ref}) \times 100$, overbar denotes the regional average), and 4) centered RMSE, which is a RMSE calculated after the regional averages are removed.

favorable for producing MCSs identified from the observed MCS database but at lower frequency, leading to underestimation of MCS number. They conclude that the incorrect response of moist (deep convection) parameterizations to the large-scale environment is likely the main reason for the bias. Pryor et al. (2020) compared the present-day and future simulations by WRF and CAM-MPAS in terms of the mean annual energy density from the wind turbines derived from the near-surface wind speed,



noting significantly weaker near-surface winds in CAM-MPAS than WRF. Their sensitivity test indicates the overestimated drag from the turbulent mountain stress parameterization in CAM5, also reported by Lindvall et al. (2013). Because model biases are largely inherited from the CAM5.4 parameterizations, previous studies using the CAM5 physics with VR approach are also useful to understand the model behavior (Huang et al., 2016; Rhoades et al., 2016, 2018b; Gettelman et al., 2018).

**Table F1.** Climatological means of selected variables over CONUS from present-day (eval) simulations.

| Variable | UR240 | UR120 | VR50-200 | VR25-100 | VR12-46 |
|---|---|---|---|---|---|
| sfc. air temperature (K) | 288.1 | 287.73 | 287.52 | 287.34 | 288.06 |
| precipitation (mm d$^{-1}$) | 1.91 | 2.03 | 2.04 | 2.10 | 2.16 |
| convective precip. (mm d$^{-1}$) | 0.89 | 0.86 | 0.88 | 0.88 | 1.01 |
| large-scale precip. (mm d$^{-1}$) | 1.02 | 1.17 | 1.16 | 1.22 | 1.15 |
| precipitable water (kg m$^{-2}$) | 21.16 | 20.12 | 19.69 | 19.34 | 19.51 |
| column cloud liquid (g m$^{-2}$) | 59.73 | 60.00 | 53.70 | 51.06 | 37.37 |
| column cloud ice (g m$^{-2}$) | 29.49 | 27.84 | 22.97 | 20.65 | 18.19 |
| total cloud fraction (fraction) | 0.52 | 0.50 | 0.48 | 0.46 | 0.41 |
| relative humidity at 850 hPa (%) | 57.73 | 56.73 | 54.56 | 53.84 | 52.16 |

**Table F2.** Examples of number of grid columns in the regional model simulations from the NA-CORDEX and FACETS archives

| Models | 50km | 25km | 12km |
|---|---|---|---|
| RegCM4 | 30429 | 123825 | 310761 |
| WRF | 24009 | 96036 | 255000 |