# Peer review of "Technical descriptions of the experimental dynamical downscaling simulations over North America by the CAM5.4-MPAS4.0 variable-resolution model"

_EGUsphere, 2022_

## Referee Comment (RC1)

The paper of **Sakaguchi et al.**, entitled '*Technical descriptions of the experimental dynamical downscaling simulations over North America by the CAM5.4-MPAS4.0 variable-resolution model*' ; https://egusphere.copernicus.org/preprints/2023/egusphere-2022-1199/ ; presents the design of the CAM-MAPS coupling and the comparison of experiments using variable resolutions (against uniform resolutions).

The paper combines technical descriptions of the experimental set-up and of the post-processing and investigation of the sensitivity to the resolution, and, additionally gives the main insights of the simulated climate for present-day and future projection (RCP 8.5 scenario).

The paper is well organized. It presents the main framework, a view of the results, but it also presents the limitations that arise to computation constraints, physical choices, … Additional materials in annexes are relevant. The way to access to the code and data is fine, with also some tools to prepare experiments, to read and/or interpolate outputs.

I only have very minor questions and suggestions that I give below. With this, I think that only a minor revision would be needed to improve the paper before accepting its publication.

--- --- --- --- ---

**Main remarks:**

*CAM-MAPS coupling:*
**Lines 170-181:** It is not easy to follow how exchanges take place along the vertical grid(s). Is there an interpolation/extrapolation along the vertical? If possible, a kind of schematic view of the way it is done could help to better understand.

**Figure 2** is the only figure that explains the MPAS-CAM coupling and I think it could be improved to better support the text:
  - Adding a box could help to identify CAM;
  - What is the role of subroutines p_d_coupling/d_p_coupling? vertical interpolation? Timestep management?
  - Are surface parameters tendencies provided to MPAS or are they only used inside CAM physics?

*Section 3.1*:
I think this section, and in particular the two first paragraphs could be reorganised for more clarity… maybe speaking first about the different resolutions used, then describing the time-stepping and the physics schemes in CAM. Finally explanations of the different runs with the "eval" and "rcp8.5" simulations could be given (and only at that final point, the specific treatment of the sea surface/sea-ice could be given, and also the explanations about the differences with downscaling experiments done with limited area models, i.e. direct downscaling from GCMs or "pseudo-warming" downscaling with addition of climate-change signals).

--- --- --- --- ---

**Other comments:**

*1. Introduction*
line 74: The reference here should be to Figure 3 I think.
Caption of Figure 1 and line 88: Maybe, it's not necessary here to speak about the use of ERA-interim SST and sea-ice cover, as this piece of information suits more to the experiments description in the section 3.

*2. Model description*
line 128: "*… the default physics option for this version of CESM…*". Two versions are cited in this

sentence. If possible, rephrase to better relate "default physics" with "version 6".

line 169: "*… the updated atmospheric and tracer states are passed to the CAM physics.*". Please see my remarks about Figure 2. As there are two groups of parametrizations, a better identification of CAM "contours" would be useful.

**3. Downscaling experiments**

line 222: "*...but still covers the most of the NA CORDEX domain.*". Not precise.

Line 274: "*climatological*"

line 297: "*...the so-called NAM grid…*". Please refer also here to Table 2 which contains information about NAM grid resolutions.

**5. Simulations**

line 390: I think it's Table 5 (and not 2).

line 478: "*CONUS*"

line 544: "*...in SAT…*" in TAS?

**Annexe C**

Tables C3 and C4: I am just wondering what are the meaning of having daily max/min temperatures in 6-hourly and 3-hourly outputs. Is it to identify the timing of these minimum and maximum in the day?

---

## Author Response (AR1)

**Replies to reviewer #1**

We thank reviewer 1 for going through all the materials including the appendices and providing us helpful suggestions. Please see below for our responses to your comments. The comments from the reviewer are shown in italics, and our responses are written in the regular font.

**Main remarks:**

**CAM-MAPS coupling:**

*Lines 170-181: It is not easy to follow how exchanges take place along the vertical grid(s). Is there an interpolation/extrapolation along the vertical? If possible, a kind of schematic view of the way it is done could help to better understand. Figure 2 is the only figure that explains the MPAS-CAM coupling and I think it could be improved to better support the text*

Thank you for the suggestions. We have revised Figure 2 (shown below) and corresponding text to make clearer the coupling between CAM and MPAS in the context of the overall process coupling within CESM1.5 (in the AMIP configuration).

[Figure]

**Revised Figure 2**

• *Adding a box could help to identify CAM;*

Done.

• *What is the role of subroutines p_d_coupling/d_p_coupling? vertical interpolation? Timestep management?*

Those two subroutines do not perform the tasks you mentioned. Realizing that more than the two subroutines were involved in all the functions necessary for the coupling, We have replaced the two subroutine names by "CAM interface code to dynamical core" in the revised figure. The role of the interface layer has been added to the main text (section 2.3 CAM-MPAS coupling), which has been reorganized with additional texts to answer your questions:

"The CESM coupler is responsible for time-step management and sequential coupling of component models. When CAM is called by the coupler, the CAM driver cycles the dynamics, physics parameterizatons, and communication with the coupler. When the dynamics is called by the CAM driver, the MPAS dynamical core receives tendencies of horizontal momentum, temperature, and mixing ratios that are predicted by physics parameterizations and the other CESM component models, and summed by the CAM driver prior to the communication with MPAS."

"... After MPAS completes its (sub) time steps, the updated atmospheric and tracer states are passed to CAM through the interface, including hydrostatic pressure, pressure thickness of each grid box, and geopotential height. The last three variables are required by the CAM physics that operates on a vertical column under hydrostatic balance, without the need to know that the vertical column is discretized in a height-based or hybrid pressure-based coordinate. No vertical interpolation or extrapolation are performed in coupling CAM and MPAS. The CAM-MPAS interface layer also calculates hydrostatic pressure velocity and performs other required conversions (e.g., convert the prognostic winds normal to cell edges to conventional u and v winds at cell centers, and mixing ratios defined with dry air in MPAS to those with moist air in CAM)."

• *Are surface parameters tendencies provided to MPAS or are they only used inside CAM physics?*

They are only used inside the CAM driver, and this point has been made clearer in the revised figure. We have mentioned the surface and other process coupling in the main text as well:

"...the MPAS dynamical core receives tendencies of horizontal momentum, temperature, and mixing ratios that are predicted by physics parameterizations and the other CESM component models, and summed by the CAM driver prior to the communication with MPAS."

*Section 3.1*:

*I think this section, and in particular the two first paragraphs could be reorganised for more clarity... maybe speaking first about the different resolutions used, then describing the time- stepping and the physics schemes in CAM. Finally explanations of the different runs with the "eval" and "rcp8.5" simulations could be given (and only at that final point, the specific treatment of the sea surface/sea-ice could be given, and also the explanations about the differences with downscaling experiments done with limited area models, i.e. direct downscaling from GCMs or "pseudo- warming" downscaling with addition of climate-change signals).*

Thank you very much for the suggestion. We have revised section 3 to address your comments, which we believe substantially improved the readability and clarity of the section.

Specifically, we have re-ordered the materials, made several minor edits throughout the section, added one more subsection, and changed the subsection titles as follows:

**3.1. Model grid and parameters**

Resolutions and model parameters (e.g., time step) used for the experiment

**3.2. Model configurations**

Overall CESM configurations (land, sea ice, etc.) used for the experiment

**3.3. Model experiments and input data**

Description of the eval and rcp8.5 simulations, input data and their preparation, and differences from limited-area modes in some aspects of the CAM-MPAS downscaling experiments.
* * *
**Other comments*:**

*1. Introduction*

*line 74: The reference here should be to Figure 3 I think.*

Thanks, we have added a reference to Figure 3.

*Caption of Figure 1 and line 88: Maybe, it's not necessary here to speak about the use of ERA- interim SST and sea-ice cover, as this piece of information suits more to the experiments description in the section 3.*

Agreed and have removed sentences referring to ERA–Interim SST.

*2. Model description*

*line 128: "… the default physics option for this version of CESM…". Two versions are cited in this sentence. If possible, rephrase to better relate "default physics" with "version 6".*

We have revised the sentences as follows

"The atmospheric component model CAM has multiple versions of physics parameterization package. We use the CAM version 5.4, which is an interim version toward CAM version 6 (Bogenschutz et al., 2018). The CAM5.4 physics is the default option for CAM in CESM1.5."

*line 169: "… the updated atmospheric and tracer states are passed to the CAM physics.". Please see my remarks about Figure 2. As there are two groups of parametrizations, a better identification of CAM "contours" would be useful.*

Yes, a contour grouping the CAM components has been added to the figure.

**3. Downscaling experiments**

*line 222: "...but still covers the most of the NA CORDEX domain.". Not precise.*

We have revised it as "... but still covers the most of North America."

*Line 274: "climatological"*

The typo has been Fixed.

*line 297: "...the so-called NAM grid...". Please refer also here to Table 2 which contains information about NAM grid resolutions.*

Thanks, we have added a reference to Table 2.

**5. Simulations**

*line 390: I think it's Table 5 (and not 2).*

Thank you, we have revised the reference.

*line 478: "CONUS"*

Thank you, the typo has been fixed.

*line 544: "...in SAT..." in TAS?*

Yes, it is. We have fixed the typo.

**Annexe C**

*Tables C3 and C4: I am just wondering what are the meaning of having daily max/min temperatures in 6-hourly and 3-hourly outputs. Is it to identify the timing of these minimum and maximum in the day?*

That is right, these were mistakes. We realized that most of the variables listed in Table C3 and Table C4 were either wrong or missing; we apologize for these mistakes and have completed those two tables with correct information.

We also have corrected the units in the tables using the exponential format following the GMD guideline.

We appreciate reviewer #1 for going through the tables and letting us know the mistakes.

**Replies to reviewer #2**

We thank reviewer 2 for going through our manuscript, in particular for evaluating our manuscript from the potential data user's point of view. We have incorporated your suggestions in our revision to make readers aware of additional aspects of our experimental protocol that need to be resolved in future studies. Please see below for our responses to your comments. The comments from the reviewer are shown in italics, and our responses are written in the regular font.

**Main remarks:**

*Reading with an eye to how one might use these simulations, I feel there are several issues with the experimental protocol that are not adequately addressed. These issues may be better explored in subsequent research, but I feel they need to be raised here. Primarily, there are many elements being varied among the simulations compared so it is difficult to assess which is responsible for the results. A couple examples:*

1. *The global precipitation increases with model resolution is not intuitive and is made more complex do to tuning of the convective scheme.*

That is probably the case, but our intention here is to explore viable options to mitigate known resolution sensitivity of the CAM hydrological cycle, and to share the result (both success and failure) with the community. To adhere more closely to this goal, we have revised the following sentences describing the sensitivity of global-mean precipitation in section 5.2.1 "Present-day climate":

"... This unexpected resolution sensitivity is not necessarily an improvement for the model hydrological cycle, and attributed to the changes we made in the convective time scale of the ZM convection scheme (Sect. 2.2) based on our previous study (Gross et al., 2018). It would be more preferable that the total precipitation and fractions of convective (associated with unresolved updraft) and large-scale (associated with resolved upward motion) components remain unchanged for grid resolution coarser than the so-called "gray-zone" (e.g., Fowler et al., 2016). However, our result does illustrate a potential (and cursory) use of the convective time scale for tuning CAM-MPAS VR simulations. For example, smaller changes than we made in the time scale (Table 3) may result in more preferable partitioning of precipitation components. Readers are referred to section 8b of Gross et al. (2018) for in-depth discussion about tuning mass-flux based convection parameterizations for VR models. "

2. *The delta SST is taken from MPI and may not be consistent with the circulation shift seen in the free-running CAM model.*

Thank you for pointing this out. We have added the following to section 3.3 "Model experiments and input data":

"... Also of note is that the SST or near-surface air temperature (TAS) biases and their changes in the MPI simulations differ from those of fully coupled CESM simulations with CAM5 or CAM6 (Meehl et al., 2012, 2013; Danabasoglu et al., 2020). Specifically, our CAM-MPAS downscaling data describe the response of the atmosphere to the ocean conditions derived from the external data (as with the case for regional model simulations in NA-CORDEX), which may be very different from the climate evolution simulated by CAM-MPAS being coupled to an active ocean model. Because CAM-MPAS and other VR atmosphere models are typically a part of global coupled climate models, it is possible (but beyond our scope) to use ocean boundary conditions derived from fully coupled simulations of the host model, or to run a fully coupled VR simulation, which provide climate-change signals that have co-evolved with the same atmosphere model."

**Reply to the editorial office's recommendation**

We thank the editor and editorial team for handling our manuscript and review process. Considering the recommendation about the color schemes (avoid using red and green colors together), we have changed colormaps for Figures 4, 8, 10, 11, 12, and D1, for which we think the color scheme is important to identify the features discussed in the text.

Best regards,

Koichi Sakaguchi

---

## Author Response (AR2)

**Reply to the editor's recommendations**

We deeply thank the editor for carefully reviewing our revised manuscript. Please see below for out point-to-point reply to the comments. The comments from the editor are shown in italics, and our responses are written in the regular font.

*Thank you for this revised manuscript that addresses the remarks of the two referees. I would like to mention however that the document egusphere-2022-1199-ATC3.pdf does not highlite all the modifications you made in the text and this made my review work somewhat more difficult to achieve. Before publication, I would like you to consider the following questions and remarks :*

I apologize for the inconsistencies in the track-change version, making you spend longer on our manuscript. The main reason for the inconsistencies is that I have not been able to find a comprehensive tool to produce track-changed manuscript in pdf format (not the latex source code) from multiple latex files in an project created at the Ovealeaf website, an on-line collaborative latex editor that I used.

After discussing with co-authors and colleagues, I decided to manually color and strike-through for each change to create a track-changed version of the manuscript pdf file, and then remove all the latex commands to produce the final version without track changes. Due to the risks to introduce errors between the track-changed and final versions, I only highlighted major changes in my track-changed version.

The "latexdiff" tool is recommended in the GMD submission webpage, but this software does not straightforwardly work with a project with multiple tex files in a modular structure. Several approaches have been recommended to enhance latexdiff on this aspect, but I could not make them work with my files (e.g., http://www.brechtdeman.com/blog/LaTeX-diff.html).

The Ovealeaf website introduces two latex packages to produce track-changed manuscript file: TrackChanges and Changes packages. Both require manually enclose sentences to be edited or removed by certain commands (e.g., \add{}). TrackChanges does not allow using any latex commands, such as referring papers or figures (e.g., \ref{}), and multiple sentences within the track-change command, thus limiting its use. The Changes package does not have a so-called "final" option to accept all the changes at once, therefore the workflow will be just the same as using the latex-native coloring and strike-through commands.

I'd really appreciate if GMD editorial team could point me to a tool or workflow that enables one to create track-changed manuscript files more easily from a modular-style latex/overleaf project.

- *L87 : consider adding « in » in « and ERA-Interim »*

Yes, it is done.

- *L288 and L290, for example, and also in Figure 4 ; sometimes you refer to the MPI model as « MPI-ESM-LR » and sometimes as « MPI model » and sometimes as « MPI » ; please make this more uniform across the text*

All references to the MPI model have been changed to MPI-ESM-LR, including the subplot titles in Figures 10, 11, and 12. We have also noted that the model name abbreviation "MPI-ESM-LR" first appears on page 9 around L200 instead of page 12 ~L260 where we introduced the MPI-ESM-LR model and its reference (Giorgetta et al., 2013). Accordingly, we have moved the first introduction of the abbreviation and model reference up to page 9.

- *In Table 4, you sometimes use « remapped to » (as in « VR50-200 remapped to NAM-44i ») and sometimes not (as in « VR25-100 NAM-22i, 1991-1995 » that should be « VR25-100 remapped to NAM-22i, 1991-1995 » I suppose) ; please make this uniform. I think just using « to » would be fine, e.g. « VR50-200 to NAM-44i » and « VR25-100 to NAM-22i, 1991-1995 »*

Yes, that's what we meant. Thank you for the suggestion. We have shortened "remapped to" to "to", and have added "to" to all the three rows you mentioned.

- *In Table 4, I don't understand the lines « VR25-100 NAM-22i to original » and « WRF 25km NAM-22i to original » mean or represent ; maybe I have missed some explanations in the text ? Please provide more details.*

Sorry for the confusion. Those lines refer to the results after remapping the precipitation fields twice: once from the original MPAS grid (VR25-100) to the NAM-22i grid, then back to the MPAS grid, on which the statistics are calculated. The row headers for these lines and the table caption have been revised for better clarification.

- *Lines 356-358 : you write that Table 4 compares different remapping methods ; this is not right I think as it is mentioned in Table 4 captions that the first-order conserve remapping method is used. It is Figure 5 that compares different remapping methods. Please correct.*

Thank you, the mistake has been corrected so that the sentence refers to Figure 5 instead of Table 4.

Best regards,

Koichi Sakaguchi